

# Linking canopy reflectance to crop structure and photosynthesis to capture and interpret spatiotemporal dimensions of per-field photosynthetic productivity

Wei Xue[1,*], Seungtaek Jeong[1], Jonghan Ko[1,*], John Tenhunen[2]

[1]Department of Applied Plant Science, Chonnam National University, 500757 Gwangju, South Korea

[2]Department of Plant Ecology, BayCEER, University of Bayreuth, 95440 Bayreuth, Germany

[*]Corresponding author: xuewei8341@jnu.ac.kr; jonghan.ko@jnu.ac.kr. Tel. 82-62-530-0753; Youngbong-ro 77, Buk-gu, Gwangju 61186, South Korea

**Abstract**. Nitrogen and water availability are two of staple environmental elements in agroecosystems that can substantially alter canopy structure and physiology then crop growth, yielding large impacts on ecosystem regulating/production provisions. However, to date, explicitly quantifying such impacts remains challenging partially due to lack of adequate methodology to capture spatial dimensions of ecosystem changes associated with nitrogen and water effects. A data assimilation, where close-range remote sensing measurements of vegetation indices derived from a hand-held instrument and an unmanned aerial vehicle (UAV) system are linked to leaf and canopy photosynthetic traits quantified at plot level by portable chamber systems, was applied to capture and interpret inter- and intra-field variations in gross primary productivity (GPP) in lowland rice grown under flooded condition (paddy rice, PD) subject to three available nutrient availability and under rainfed condition (RF) in East-Asian monsoon region, South Korea. Spatial variations (SVs) in both GPP and light use efficiency ($LUE_{cabs}$) early in growing season were amplified by nitrogen addition, and such nutritional effects narrowed over time. Shift planting culture from flooded to rainfed conditions strengthened SVs in GPP and $LUE_{cabs}$. Intervention of prolonged drought event at late growing season dramatically intensified their SVs that are supposed to seasonally decrease. Nevertheless, nitrogen addition effects on SV of $LUE_{cabs}$ at early growth stage made PD field exert greater SVs than RF field. SV of GPP across PD and RF rice were likely related to LAI development less to $LUE_{cabs}$ while, numerical analysis suggested that consider spatial variation and strength in $LUE_{cabs}$ for the same crop type tends to be vital for better evaluation in landscape/regional patterns of ecosystem photosynthetic productivity at critical phenology stages.

**Key words:** photosynthesis, remote sensing, rice, spatial variation, UAV

## 1 Introduction

Agricultural landscape in most Asia monsoon regions is featured by multicultural cropping systems comprising of relatively small land holdings under 2 ha (Devendra, 2007). Changes in phenology of those crop ecosystems where rice makes up larger portion and exerts a rapid completion of life cycle in a short period of time with markedly changes in canopy



dynamics are of significant importance in regional controls of carbon balance and biogeochemical processes (Kwon et al., 2010; Lindner et al., 2015; Xue et al., 2017), tending to be one of drivers causing seasonal fluctuations of atmospheric $CO_2$ concentration in north hemisphere (Forkel et al., 2016). Hence, to better understand their ecological implications under current climate and environmental changes, one of main concerns lies in spatiotemporal aspects of ecosystem photosynthetic

productivity in the staple crop subject to different methods of field management and anthropogenic interventions, and underlying physiological mechanisms that are responsible for such spatiotemporal dimensions.

    The stability, repeat measurement capability, and landscape to global coverage of remote sensing from satellites have triggered widespread use of such measurements to obtain spatial patterns of biophysical and biochemical variables in studies of land surface and atmospheric process (Richardson et al., 2013). Recent study introducing satellite products as input

parameters in flux modelling campaigns carried out in small size of crop land reported that prediction accuracy seems to be pixel-size dependent (Adiku et al., 2006), yielding better prediction if apply satellite products at finer resolution. Accordingly, attempts made to assimilate those parameters into process-based crop growth models that led to noticeable overestimations and/or underestimations in plant functional traits over a whole growing season have been increasingly concerned (Tenhunen et al., 2009; Lee, 2014; Alton, 2017). Satellite images collected during plant growing seasons have been used to monitor crop

growth and to predict yield production, but their use has been limited by poor revisit times, coarse spatial resolution, and/or cloudy weather. They technically conceal delicate fluctuations of ecosystem productivity tightly associated with per-field ecological conditions on which plants survival and dispersal depend (Seo et al., 2014), and hence bring great spatiotemporal uncertainties in evaluating strength of daily carbon fluxes among micro sites of the same plant function type at principle growth stages. The research gaps might be mathematically resolved using complex Bayesian melding (Gelfand, 2012).

Multi-pragmatic solutions are suggested to develop spatial/temporal data fusions that integrate spatially hierarchical remote sensing networks and *in situ* ground surface observations (Lausch et al., 2016; Pause et al., 2016), aiming to better monitor canopy dynamics and environmental impacts on them.

    Of them that help to understand per-field ecological processes, close-range remote sensing technique rises to be one realistically convenient measure that can timely provide us with temporal information of ecosystem dynamics at high spatial

resolution. Recent applications in agronomy studies (Zhang and Kovacs, 2012; Ko et al., 2015; Jeong et al., 2016) refract the feasibility of resolving the research gaps in terms of capturing spatiotemporal aspects of ecosystem photosynthetic productivity at intra- and inter-fields.

    To well interpret spatiotemporal variations of ecosystem photosynthetic productivity captured by close-range remoter sensing, conventional physiological studies at canopy leaves are, nevertheless, essential (Sinclair and Horie, 1989; Niinemets

and Tenhunen, 1997). As leaves are the small and basic units that constitute rice canopy volume, their functioning could change with canopy development and changing habitat conditions (Xue et al., 2016a, b), contributing to fluctuations in strength of seasonal canopy photosynthesis.

Traditional ecophysiology approaches are greatly limited to compare neighboring plant individuals and tend to neglect spatial dimensions. Landscape ecology, although resolving ecosystem functioning at broader scale, is commonly restricted to regional analysis at higher hierarchical level beyond individual organisms. Therefore, the central aims of this research are to construct a spatially integrative concept model that assimilates quantitatively abundant data sets collected from a close-range

remote sensing system applied at field level and from traditional ecophysiology approaches at plot level, and capture and then interpret effects of different field management practices i.e. nutrient application and water treatments on temporal and spatial aspects of ecosystem photosynthetic productivity via their influences on canopy leaf physiology and structure, to evaluate the following hypothesis:

(1) Temporal course of canopy carbon gain capacity was primarily driven by LAI development and solar radiation intensity

at reproductive stage (Xue et al., 2016a; 2017). Nevertheless, canopy leaf physiology is one of primary factors that determine canopy light use efficiency and thereby carbon gain capacity (Sinclair and Horie, 1989). Hence, spatial variability of ecosystem GPP could be concurrently driven by canopy structure i.e. LAI and canopy leaf physiology i.e. $LUE_{cabs}$.

(2) Shifts of planting culture from flooded to rainfed conditions mean that water availability tends to be a primary factor determining ecosystem photosynthetic productivity, and then growth of rainfed rice suffers from multiple uncertainties

regarding timing/strength of precipitation and uptake of nutrient availability in soil (Kato et al., 2016). Significant changes in leaf and root anatomies, and canopy structure and function in rainfed field could occur (Yoshida, 1981; Steudle, 2000). Greater variations in spatial aspects of ecosystem GPP, LAI and $LUE_{cabs}$ in rainfed lowland rice than flooded rice are therefore anticipated.

**2 Materials and Methods**

**2.1 Study site**

Field campaign was carried out at the agricultural field station of Chonnam National University, Gwangju, S. Korea ($35^o10'$N, $126^o53'$E, altitude of 33 m, Fig. 1). Mean annual air temperature and precipitation averaged over past two decades are approx. $13.8^oC$ and 1400 mm $yr^{-1}$. East-Asian monsoon climate is prevalent from June to October in this region during

which time more than half of annual precipitation fall. The top layer of soil is categorized as loam with sand of 388 g $kg^{-1}$, silt of 378 g $kg^{-1}$, clay of 234 g $kg^{-1}$, PH of 5.5, organic C content of 12.3 g $kg^{-1}$, available P of 13.1 mg $P_2O_5$ $kg^{-1}$, and total N before fertilization of 1.0 g $kg^{-1}$. Thirty-day-old seedlings of a newly breeding line *Oryza sativa* cv. Unkwang (Kim et al., 2006) were transplanted into flooded fields named paddy rice (PD) on May 20, 2013 (140 days of year, DOY). N:P:K with mass ratio of 11:5:6 was mixed to generate three fertilizer application rates: 0 kg N $ha^{-1}$ (no supplementary fertilizer, plot size

~511 $m^2$, named low nutrient group), 115 kg N $ha^{-1}$ (plot size ~1387 $m^2$, normal nutrient group), and 180 kg N $ha^{-1}$ (plot size ~511 $m^2$, high nutrient group) (Fig. 1). Nutrient treatment groups were respectively isolated by 35 cm width perimeter cement walls, inserted into the soil 1 m depth. 80% of total nitrogen fertilizer was applied by hand spreading two days before

transplanting, and the rest used at active tillering phase of vegetative stage. P fertilizer was applied as 100% basal dosage, and K fertilizer was applied as 65% basal dosage and 35% during tillering phase. Seeds of the same rice cultivar were directly sown in an adjacent upland field, being treated as rainfed rice (RF, ~64 m$^2$) on April 22 (112 DOY). The same fertilizer compound with 115 kg N ha$^{-1}$ as PD normal nutrient group was conducted in RF field two times, 80% before

seeding and the rest at tillering phase. No irrigation was supplied at the RF field during the whole growing season. All field management practices conformed to local planting cultures. Life history in Unkwang rice generally aligned to a classification of phenology in temperate rice proposed by Yoshida (1981) that spends about 30 days in the vegetative stage after transplanting, 30 days in the reproductive stage, and 30 days in the ripening period.

To better underpin physiological mechanisms that may contribute to spatial patters of per-field photosynthetic

productivity, a pair of experiment consisting of PD and RF Unkwang rice in a controlled growth chamber at University of Bayreuth (11$^o$34′N, 49$^o$56′E) was deployed in September 2014. 30-day-old seedlings were transplanted into plastic containers (top diameter 25.4 cm and height 25 cm) with similar plant spacing as planting practice in the 2013 field experiment. The equivalent fertilizer 115 kg N ha$^{-1}$ was applied two times in both PD and RF rice, before transplanting/sowing and at tillering phase. All plants were then acclimated in the growth chamber to daytime air

temperature 30$^o$C, relative humidity 60%, night temperature of 25$^o$C, and light intensity of 900 μmol m$^{-2}$ s$^{-1}$ (35.64 MJ m$^{-2}$ d$^{-1}$). Soil water content (SWC) in RF rice containers was maintained between 0.2 and 0.4 m$^3$ m$^{-3}$ using EC-5 soil moisture sensors (EC-5, Decagon, WA, USA).

### 2.2 Field measurements of meteorological factors and soil water content

Meteorological factors including air temperature, relative humidity, wind speed, precipitation, and global radiation were continuously measured with a 2 m height automatic weather station installed at a field margin of RF field (AWS, WS-GP1, Delta-T Devices Ltd., UK). Weather data were recorded every 5 min, averaged and logged half-hourly. Additionally, values of SWC at 10, 30 and 60 cm depth at three sites in RF field were continuously measured every 15 min using EC-5 soil moisture sensors. SWC data recorded by the EC-5 sensors were then calibrated by actual SWC measurements conducted in

the laboratory with the same soil. SWC was then converted to soil water potential ($\psi_s$) with standard soil-water retention curves of Van Genuchten (1980), referred to Xue et al. (2016b).

### 2.3 Field measurements of diurnal courses of leaf and canopy $CO_2$ exchange

Diurnal gas exchange and chlorophyll fluorescence measurements in fully expanded uppermost, second, third and fourth

leaves of canopy profiles at PD high nutrient group were conducted on 57 and 73 day after transplanting (DAT) (197 and 213 DOY) using a portable gas exchange and chlorophyll fluorescence system (GFS-3000 and PAM Fluorometer 3050-F, Heinz Walz GmbH, Effeltrich, Germany) to track ambient environmental conditions external to leaf cuvette. Repeated





measurements of diurnal course of leaf gas exchange were carried out in uppermost leaves in PD low nutrient group on 171, 172, 179, 180 and 199 DOY (31, 32, 39, 40 and 59 DAT), in PD normal nutrient group on 175, 177, 195 and 211 DOY (35, 37, 55, and 71 DAT), in PD high nutrient group on 170 and 178 DOY (30 and 38 DAT), and in RF rice on 157, 181, 201, 205, 222, 223, 227, 231, 235 and 238 DOY. Mid parts of two or three leaves were enclosed into the leaf chamber from sunrise to

sunset. Photosynthetic rate and momentary micrometeorological factors just above plant canopies were recorded every 5 min, and automatic calibration executed by a user-defined program was repeated every 15 min. Leaf light use efficiency based on incident PAR (LUE$_{leaf}$) was estimated using photosynthesis data recorded at incident PAR less than 200 μmol m$^{-2}$ s$^{-1}$.

Diurnal course of canopy gas exchange was conducted by a custom-built transparent chamber (L 39.5 × W 39.5 × H 50.5 cm) used for net ecosystem gas exchange (NEE) measurement and by a opaque chamber (L 39.5 × W 39.5 × H 50.5 cm)

designed for ecosystem respiration (R$_{eco}$) measurement (Lindner et al., 2016; Xue et al., 2016a) on ~ 159, 167, 175, 200, 220, and 240 DOY. Measurements on 240 DOY were only available at PD normal group and RF rice. Four white frames, with three filled with healthy plants and one set on bare soil without any plants, were deployed in each PD nutrient group and in RF field. They were inserted into the soil at the 10 cm depth before transplanting/sowing to block air leak at the interface between the frame and soil surface, and kept in the fields until plants were harvested. Diurnal courses of NEE and R$_{eco}$ per

square meter were monitored at hourly intervals from sunrise to sunset. Differences of air temperature between inside and outside the chamber were controlled less than 1$^{o}$C using ice packs positioned at the back side of the chamber to avoid shadow effects of ice packs. Incident PAR inside the transparent chamber was measured with a quantum sensor (LI-190, LI-CPR, Lincoln, Nebraska, USA). GPP estimation was derived by,

$$GPP = -NEE + R_{eco} \qquad\qquad (1)$$

where R$_{eco}$ rates at times when NEE rates were measured were determined from an exponential regression with respect to chamber air temperature (T$_{air}$). A classical hyperbolic light response function was fit to estimate gross primary productivity (GPP, sum of NEE and R$_{eco}$), yielding canopy light use efficiency (LUE$_{cint}$) defined as the initial slope of the response and an estimate of maximum GPP rate (GPP$_{max}$) at relatively infinite high PAR level.

**2.4 Field measurements of canopy reflectance**

Reflectance measurements were carried out with a hand-held multispectral radiometer (Cropscan, MSR4 with 4 wave bands, Cropscan Inc., Rochester, MN, USA). Incident radiation was measured with a view-angle of 180$^{o}$, and that reflected by rice canopies was measured with a view angle of 28$^{o}$. Weekly reflectance measurement arranged around plants sampled for canopy gas exchange was repeated six times in each PD nutrient treatment and three times in RF field at solar noon midday

when sky was clear without clouds. Normalized difference vegetation index (NDVI) was a product of differences of reflectance in the field of which red (the central band-width of 660.9 nm) and near infrared (the central band-width of 813.2 nm). Estimations of ground-based NDVI were made on the days when canopy gas exchange measurements, referred to Xue



et al. (2016a).

Spectral reflectance at fine spatial resolution less than 10 cm for the whole PD field and RF field was measured on July 21 (172 DOY, vegetative stage), July 11 (192 DOY, early reproductive stage), July 25 (206 DOY, middle reproductive stage), August 08 (220 DOY, early ripening stage), and August 21 (233 DOY, middle ripening stage) using an unmanned aerial

vehicle (UAV) system (detailed construction of the UAV system referred to Jeong et al. (2016)). The UAV images were acquired at approximately local noon ± 30 min (i.e. KST 12:10 to 13:10) when there were clear skies or homogenous cloudy skies. The camera exposure was set at its minimum value ($0.5\ \mu m\ s^{-1}$) under clear sky conditions and ranged between 1.0 to 2.0 μm/s under homogenous fine cloudy skies to obtain the best images. When recording UAV images, the multispectral camera (mini-MCA6, Tetracam Inc., Chatsworth, California, USA) loaded on board the UAV which detected ground

reflectance with the wavelength bands of 450, 550, 650, 800, 830, and 880 nm was always positioned vertically to the ground.

Pseudo invariant targets (PITs) at three different colors (white, black, and gray) were placed adjacent to PD field prior to each UAV flight. At-surface reflectance values of two selected waveband at 800 and 650 nm from those PITs were obtained using the other hand-held spectrometer (Cropscan, MSR16 with 16 wave bands). Linear regression correlations

were made between mini-MCA6 digital values and the reflectance from the MSR16 at each corresponding waveband, with correlation coefficient ranging from 0.98 to 0.99 (descriptions in detail referred to Ko et al. (2015) and Jeong et al. (2016)). Camera measurements were then calibrated based on at-surface measurements by applying each linear regression to the field imagery. Evaluation of the radiometric corrected UAV images was carried out by comparisons with measurements of sixteen ground point reflectance values which comprised 12 points in paddy fields, 4 points in bright cement, dark asphalt, bare soil,

and tilled soil. There were close correspondences between reflectance derived from the radiometric corrected UAV images and those measured at the ground over all UAV flight dates, with correction efficiency (E) up to 0.99 and root mean square error (RMSE) ranging between 0.01 and 0.05 (Appendices, Fig. A1). Radiometric calibrated reflectance at red, green, and blue bands (450, 550, and 650 nm) on June 21/172 DOY (clear sky) when there had low density vegetation canopies with large exposure of water surface were consistently lower than at-surface measurements (Appendices, Fig. A1a), resulting in

risks of overestimating field NDVI (a product of differences in reflectance of red 650 nm and near infrared 800 nm) thereby biased estimation of $GPP_{day}$ and $LUE_{cabs}$. For sake of brevity the radiometric calibrated camera reflectance of red waveband on June 21/172 DOY were recalibrated by a linear regression line against at-surface measurements (Appendices, Fig. A1a, $\rho_{red\_ground\ meas.} = 1.761 * \rho_{red\_UAV}$, $R^2 = 0.76$, $p < 0.01$).

**2.5 Measurements of leaf area, nitrogen content and leaf water potential**

After conducting leaf and canopy gas exchange measurements, leaf samples were collected to estimate leaf area and nitrogen content. Three bundles consisting of fifteen plants from each treatment were harvested on 26, 33, 54, 72 and 86 DAT, and





total plant area (leaf and stem) was determined with an LI-3100 leaf area meter (LI-3100, LI-COR, Lincoln, Nebraska, USA).

Leaves of PD and RF rice grown in the growth chamber were harvested on 33 and 55 DAT. All plant materials were dried at

~60°C for at least two days before measurements of leaf nitrogen content. Leaf nitrogen content was quantified using a C:N

analyzer (Model 1500, Carlo Erba Instruments, Milan, Italy). Weekly measurements of LAI were conducted before 220

DOY using a portable plant canopy analyzer (LI-2000, LI-COR, Lincoln, Nebraska, USA) at the same locations where

at-surface canopy reflectance values were sampled using the Cropscan, and then these were calibrated using those by harvest

method. LAI measurements on 240 DOY were supplemented referring to Lindner et al. (2016). On the same measuring times

as leaf gas exchange conducted in August, daily courses of leaf water potential in RF rice were collected with a pressure

chamber (PMS Instruments, Corvallis, USA). Healthy and well-expanded leaves in plant canopies were enclosed in a plastic

bag before cutting and rapidly transferred into a pressure chamber.

**2.6 Data assimilation process**

Assessment of influences of field management practices i.e. nutrient and water availability in crop photosynthetic traits and

interpretation of the presence of such spatiotemporal fluctuations require development of a data assimilation process, which

could be capable of linking *in situ* observations of leaf and canopy photosynthetic traits and vegetation information at field

level. Here, a simple concept model aiming to resolve the objective stated above was developed, up-scaling application of

the classical light response model of leaf photosynthesis to canopy and field dimensions using hyperspectral reflectance of

ground surface collected at corresponding scales in the following Eqs 2-8:

$$LUE_{c\text{int}} = a_1 \times LAI + b_1 \tag{2}$$

$$GPP_{max} = a_2 \times LAI + b_2 \tag{3}$$

$$LAI = a_3 \times NDVI^2 + b_3 \times NDVI + c_3 \tag{4}$$

$$GPP_{day} = \sum_{j=1}^{N} \frac{LUE_{c\text{int}} \times GPP_{max} \times PAR_j}{LUE_{c\text{int}} \times PAR_j + GPP_{max}} \tag{5}$$

$$fAPAR = fAPAR_{max}\left[1 - \left(\frac{NDVI_{max} - NDVI}{NDVI_{max} - NDVI_{min}}\right)^{\varepsilon}\right] \tag{6}$$

$$fAPAR = a_4 \times NDVI + b_4 \tag{7}$$

$$LUE_{cabs} = \frac{GPP_{day}}{fAPAR \times PAR_{day}} \tag{8}$$

where in Eq. 2, $a_1$ and $b_1$ are regression coefficients for LUE$_{cint}$-LAI correlation based on plot measurements (Table 1). In Eq.

3, $a_2$ and $b_2$ are regression coefficients for GPP$_{max}$-LAI correlation based on plot measurements (Table 1). In Eq. 4, $a_3$, $b_3$, and





$c_3$ are regression coefficients for LAI-NDVI mathematic correlation across all data sets based on plot measurements (Table 1), which was in line with a 3-year-report in rice in terms of LAI-NDVI trajectory by Jo et al. (2015). In Eq. 5 $GPP_{day}$ is daily integrated GPP per pixel, a product of light use efficiency based on incident PAR ($LUE_{cint}$), maximum GPP rate ($GPP_{max}$) and half-hourly averaged $PAR_j$ obtained from the AWS. N is number of observations of incident PAR during daytime. In Eq. 6,

$fAPAR_{max}$, $NDVI_{max}$, $NDVI_{min}$, and ε are maximum fraction of absorbed photosynthetically active radiation, maximum NDVI and minimum NDVI of fAPAR-NDVI correlation and its coefficient in green crop canopies, referring to Table 1 and Xue et al. (2016a). $a_4$ and $b_4$ in Eq. 7 are regression coefficients for fAPAR-NDVI correlation in senescing canopies (Table 1, here refer to the stage after middle ripening stage in rice), derived from Inoue et al. (2008). Light use efficiency based on daily canopy light interception per pixel ($LUE_{cabs}$) in Eq. 8 is a product of $GPP_{day}$, fAPAR and $PAR_{day}$ (daily integrated

incident PAR).

### 2.7 Geospatial statistic

Regionalized variable theory takes the differences between pairs of values separated by a certain quantity, usually distance, commonly expressed as variance (Vieira et al., 1983). A widely used geostatistical analysis to depict the spatial correlation

structure of observations in space such as field soil fertility and temperature as well as other ecological processes is semi-variogram (Pierson and Wight, 1991; Loescher et al., 2014), given by:

$$\gamma(h) = \frac{1}{2N(h)} \sum_{j=1}^{N(h)} \left[ z(x_j) - z(x_j + h) \right]^2 \tag{9}$$

$$CV_{sill} = \frac{\sqrt{2 \times \gamma_{sill}}}{Mean} \tag{10}$$

where $z(x_j)$, j=1, 2, …, n denotes the set of $GPP_{day}/LUE_{cabs}$ data; $x_j$ is the vector of spatial coordinates of the jth observation;

h is the pixel distance of sample values (lag); N(h) is number of pairs of values separated by lag, and γ(h) is semi-variance for the lag. $CV_{sill}$ is coefficient of variance using the sill and value of the mean for estimation. The semi-variogram simply describes how the variance of observations changes with the distance in a given direction or it is averaged over all directions. The averaged semi-variance over all directions was used in this research. Most often, semi-variance values increase until they reach a maximum approximately equal to the sample variance of the measured variable known as the "sill". The lag at

which the sill is reached is known as the "range". Beyond the range, values of observations are no longer spatially correlated. Sill values refract magnitude of spatial variability of variables in the field. Several simple functions are commonly used to model semi-variogram, which must be proven to be positive definite. An exponential rise to maximum function to approximate a spherical model was used to extrapolate the value of the sill, listed below:

$$\gamma(h) = a \times (1 - \exp(-b \times h)) \tag{11}$$

where b is the sill and a is the nugget value.

### 2.8 Statistical analysis

Descriptive statistics of the data included computation of the sample mean, maximum (max.), and coefficient of variation

($CV_{traditional}$). Nonlinear least square method for GPP/PAR curves was executed using R software (R 3.2.3, R Development

Core Team, Austria). The data assimilation that links remote sensing data and ecophysiological measurements and

geostatistical analyses was processed using IDL 8.0 /ENVI 4.8 software (EXELIS Inc., Rochester, NY, USA).

### 3 Results

### 3.1 Seasonal courses of at-surface NDVI, LAI, $LUE_{cint}$, and $GPP_{max}$

ANOVA analysis for NDVI indicated that NDVI values measured around 170 DOY between the PD normal and high

nutrient groups were analogous but significantly higher than the low group at 0.05 level (Fig. 2a, p = 0.026). Statistical

difference at the significant level of 0.05 between the RF and PD low group was not found. No significant discrepancy

existed between PD normal and high groups over the growing seasons (p > 0.1). Higher NDVI at the PD fertilizer addition

groups were evident during vegetative stage and early at reproductive stage before 200 DOY (p = 0.06). Such a clear

discrepancy in NDVI between the PD low and fertilization groups and RF rice dismissed after 210 DOY (p = 0.10). NDVI

values advanced to decline after plants in the PD field arrived at maximum levels around 210 DOY. However, the RF rice

remained green around 240 DOY with higher LAI by 22.5% when plants in the PD field started senescence (Fig. 2b), which

results in relatively higher at-surface NDVI that was also captured by field image of NDVI derived from the UAV system.

LAI in the PD normal nutrient group was similar to those of the high group at the corresponding growth stages (Fig. 2b),

assembling seasonal course of NDVI for the normal/high groups. Enhanced LAI development after 180 DOY by fertilizer

addition was present, and nitrogen effects persisted until around 210 DOY, which was in line with NDVI development

among PD nutrient groups. LAI in the RF rice ranged between the PD low and fertilization groups while it remained higher

values on 240 DOY. Regression analysis for NDVI-LAI relationship in grouped datasets showed a common trajectory across

PD nutrient groups and RF rice (Fig. 3a, $R^2$ = 0.95, p < 0.001).

A curvilinear response of GPP rate to incident PAR was well fitted by the classical light response model at each

measuring date, which was previously reported (Lindner et al., 2016) and not shown here. Resulting $LUE_{cint}$ on 160 DOY

was approx. 0.01 $\mu$mol $CO_2$ $\mu$mol$^{-1}$ $PAR_{incident}$ crossing the PD nutrient groups and RF rice, and rapidly increased after 180

DOY (Fig. 2c). Differences in $LUE_{cint}$ among the PD nutrient groups were relatively small less than 20% at corresponding

dates. Nevertheless, the RF rice presented dramatically high $LUE_{cint}$ as compared to the PD rice from 180 DOY to the end of

the growing season, showing the highest values at 0.11 $\mu$mol $CO_2$ $\mu$mol$^{-1}$ and 0.05 $\mu$mol $CO_2$ $\mu$mol$^{-1}$ found in RF and PD rice,

respectively. Generally speaking, PD rice at the fertilization groups had relatively higher $GPP_{max}$ showing the maximum





level of 51.60 µmol $CO_2$ m$^{-2}$ s$^{-1}$ than the low group at 38.90 µmol $CO_2$ m$^{-2}$ s$^{-1}$ (Fig. 2d). Maximum $GPP_{max}$ in the RF rice was

analogous to that of PD rice, and remained higher on 240 DOY, which was thought to be ascribed to green LAI (Fig. 2b).

Similarities in photosynthetic traits in terms of NDVI, LAI, $GPP_{max}$ and $LUE_{cint}$ between the normal and high nutrient groups

at the corresponding growth stages were evident. Hence, comparisons in those parameters stated below were referred to the

low and normal groups.

Relatively low LAI in RF rice during reproductive stage but higher $LUE_{cint}$ than PD at the same growing stage therefore

resulted in a distinction regarding LAI-$LUE_{cint}$ correlation associated with slope (Fig. 3c, $R^2 = 0.74$, $p = 0.02$ in RF, $R^2 = 0.85$,

$p < 0.0001$ in PD, see Table 1). A common linear regression for LAI-$GPP_{max}$ correlation that interpreted 88% of variations in

$GPP_{max}$ across the PD nutrient groups and RF rice was evident (Fig. 3b, $R^2 = 0.88$, $p < 0.0001$). Canopy leaf nitrogen content

($N_m$, %) collected in both field and controlled growth chamber were significantly higher in RF rice after 180 DOY (Fig. 4a, b,

$p < 0.05$). Light use efficiency at leaf level ($LUE_{leaf}$) was positively correlated to $N_m$ (Fig. 4b, $R^2 = 0.65$, $p = 0.0007$). It

implied that improved $LUE_{cint}$ in RF rice observed after 180 DOY could be related to its strengthened capacity of nitrogen

accumulation in canopy leaves.

**3.2 Field mapping of $GPP_{day}$ and $LUE_{cabs}$**

Field maps of $GPP_{day}$ and $LUE_{cabs}$ at principle growth stages (Figs. 5 and 6) clearly showed that seasonal change of

within-field $GPP_{day}$ at each nutrient group could be quantitatively mapped using three types of colors (yellow, blue and red)

corresponding to low, medium and high numerical values. Pink pixels and bright red pixels were respectively observed in PD

and RF rice on measuring date August 08/220 DOY during which time most rice plants proceeded to ripen, showing the

highest LAI. However, colour distribution in space at specific growth stage within nutrient groups especially in normal and

low groups on July 11/192 DOY and August 21/223 DOY seems to be uneven (Fig. 5b, d). Furthermore, uneven distribution

in RF rice was intensified as compared with PD rice on corresponding dates. For $LUE_{cabs}$, appearance of greater spatial

variability in color distribution was seen at early growth stage in both PD and RF rice (Fig. 6a, e), which seems to be in

contrast with spatial aspects of $GPP_{day}$ over the growing season. $LUE_{cabs}$ distributions in space over reproductive stage (July

11/192) seemingly tend to approach homogeneities in either PD nutrient groups or RF rice (Fig. 6b, c, f, g).

Descriptive statistics including Mean, Max., and $CV_{traditional}$ in $GPP_{day}$ and $LUE_{cabs}$ described their mean, maximum

values at field scale and within-field variation of mean across the growing season (Table 2). Max. $GPP_{day}$ was differed

significantly between normal (7.29 g C m$^{-2}$ d$^{-1}$) and low (3.78 g C m$^{-2}$ d$^{-1}$) nutrient groups after four weeks after

transplantation, which was clearly indicative in visual display of pixel $GPP_{day}$ as well (Fig. 5a, d). Nevertheless, field mean

values among the three nutrient groups were close to one another. Enhanced field mean of $GPP_{day}$ in normal groups by

35.63% as compared to low group appeared on June 11/192 DOY, and the large discrepancy persisted until the end of

growing season. Except the early growth stage three nutrient groups showed similar values in maximum $GPP_{day}$ which



reached 12.49 g C m$^{-2}$ d$^{-1}$ at normal group around August 08/220 DOY and then declined at senescence stage. Maximum GPP$_{day}$ predicated using light use efficiency model in our previous report (Xue et al., 2016a) tended to be higher as compared with the one shown here at normal nutrient group, which is thought to be due to model sensitivity to changes in ambient light environment.

Rice plants grown in RF field showed significantly higher mean and maximum GPP$_{day}$ than PD rice at respective growth stages (Table 2). However, CV$_{traditional}$ in RF rice was much higher by roughly 2 times than PD normal and low nutrient groups after several weeks after transplantation. PD normal nutrient group showed the higher CV$_{traditional}$ quantified on June 21/172 DOY, then followed by high and low groups. Differences in CV$_{traditional}$ among PD nutrient groups dismissed over time, which well aligns with colour display in field map of GPP$_{day}$ in Fig. 5c and d. They imply that although fertilizer

addition in traditional way can promote increment of field average GPP$_{day}$, it dramatically strengths field variations of GPP$_{day}$ at early growth stage in paddy field. As we expect, the change in planting culture from paddy to rainfed could promote enhancement of field variations in mean of field GPP$_{day}$ probably due to rising risks in soil water availability when prolonged drought events occur.

     LUE$_{cabs}$ appeared to be higher early at the growth stage, rapidly declined after plant growth and development advanced

to reproductive stage, and gradually decreased to approx. 0.52 and 0.81 g C MJ$^{-1}$ at senescence stage in PD and RF rice, respectively (Table 2). RF rice had clearly high values of average LUE$_{cabs}$ as compared to PD by 20.93%, 35.18%, 26.43%, and 35.80% on July 11, July 25, August 08 and August 21, correspondingly, apart from June 21 during which time PD and RF showed similar LUE$_{cabs}$ around 1.4 g C MJ$^{-1}$. Enhanced LUE$_{cabs}$ in RF rice over the growing season was likely ascribed to higher leaf nitrogen content shown in Fig. 4a.

Seasonal courses of CV$_{traditional}$ of LUE$_{cabs}$ among PD nutrient groups exerted a similar tendency, assembling mean of LUE$_{cabs}$ (Table 2). CV$_{traditional}$ at normal and high nutrient groups were analogous over time while, appeared to be higher on June 21/172 DOY and July 11/192 DOY by approx. 62% and 50% than low nutrient group, respectively. Interestingly, CV$_{traditional}$ at fertilization groups (normal and high groups) exerted markedly greater values by approx. 53% and 30% than RF rice at early growth stage (June 21/172 DOY and July 11/192 DOY). Similar to drought impacts in amplifying CV$_{traditional}$

in GPP$_{day}$ on August 21/233 DOY in RF rice, amplified CV$_{traditional}$ in LUE$_{cabs}$ were observed as well. Lower CV$_{traditional}$ and similarities in LUE$_{cabs}$ over field space on July 25/206 DOY and August 08/220 DOY well corresponded to field map of LUE$_{cabs}$ at corresponding dates, meaning that field mapping in proper ways also could visibly deliver distribution information of ecosystem photosynthetic traits in space.

**3.3 Semi-variograms of GPP$_{day}$, LUE$_{cabs}$, and LAI**

Semi-variogram analysis is one of widely used geostatistical parameters to quantitatively evaluate spatial variation. Sill values were derived from exponential rise to maximum function which fits values of semi-variogram at each nutrient and/or



water treatment ($R^2 > 0.83$, $P < 0.01$). Values of $CV_{sill}$ in $GPP_{day}$ were significantly and positively correlated to $CV_{traditional}$ ($R^2 = 0.83$, $p < 0.001$, Fig. 7a), demonstrating that the semi-variogram accurately captured patterns of spatial variability in those ecophysiological traits among nutrient treatments and RF rice. Estimates of $CV_{sill}$ among nutrient groups were generally close to those of $CV_{traditional}$, approaching 1:1 line (Fig. 7a). However, $CV_{traditional}$ values in RF rice were commonly lower by

approx. 20% than $CV_{sill}$ at principle growth stages. This occurred because of the traditional method of calculating CV does not account for spatial correlation in data, implying that spatial heterogeneity in RF field associated with water availability and resulting crop growth was greater as compared to PD rice. This was also proven by average $CV_{sill}$ in RF that was greater by about 50% than that of PD rice averaged across nutrient groups (Table 3).

A significantly positive correlation between $CV_{sill}$ and $CV_{traditional}$ was observed in $LUE_{cabs}$ as well ($R^2 = 0.89$, $p < 0.001$,

Fig. 7b). All of $CV_{sill}$ sampled across PD nutrient groups and RF rice resided at right side of 1:1 line, being higher than $CV_{traditional}$ but analogous between PD and RF rice, which was different from the significant difference in $CV_{sill}$ of $GPP_{day}$ between PD and RF rice shown in Fig. 7a. It was also evident by average $CV_{sill}$ of 11.66 in RF rice that was close to 14.37 of PD rice averaged across nutrient groups (Table 3), meaning that spatial variability of $LUE_{cabs}$ in PD rice exerted great amplitude that tends to be similar to RF rice. A positively linear correlation between $CV_{sill}$ and $CV_{traditional}$ was evident in LAI

($R^2 = 0.80$, $p < 0.001$, Fig. 7c). Data points collected over PD nutrient groups oscillated closely the 1:1 line and an exception was observed in RF rice, which assembles the phenomena observed in $CV_{sill}$-$CV_{traditional}$ for $GPP_{day}$ but differs from that for $LUE_{cabs}$. Given the tight correlation between $CV_{sill}$ and sill values, sill instead of $CV_{sill}$ was used in spatial analysis for $GPP_{day}$ and $LUE_{cabs}$ as discussed below.

**3.4 Spatial patterns of $GPP_{day}$, $LUE_{cabs}$, and LAI**

Seasonal development in sill values of $GPP_{day}$ exhibited similar tendency across PD nutrient groups and RF rice that increased from vegetative stage to early reproductive stage and then declined (Table 3, upper part). Paired t-test showed that difference of sill in RF rice was significantly different from PD nutrient groups at the 0.05 level. Nevertheless, significant differences were not repeatedly observed among PD nutrient groups. At early growth seasons i.e. June 21/172 DOY

especially July 11/192 DOY, normal and high nutrient groups had relatively high sill in average by 43.90% as compared to low nutrient group, implying that fertilizer addition could contribute to spatial variability of $GPP_{day}$, which conforms to differences in $CV_{traditional}$ (Table 2). As we expect, sill of RF rice measured on August 21/233 DOY increased in contrast to observed seasonal tendency of sill that was supposed to decline, due to occurrence of a prolonged drought event from August 11 to 20 during which leaf water potential around solar noon declined down to -2.0 MPa and severe leaf rolling happened

(data not shown). Significant impacts by drought on $GPP_{day}$ were observed. Seasonal courses of sill in LAI across PD nutrient groups and RF rice were similar to those of $GPP_{day}$ (Table 3, middle part). Sills of LAI in RF rice were generally higher than PD rice at corresponding growth stages.

Sill of $LUE_{cabs}$ showed seasonal trend that was similar to $GPP_{day}$ (Table 3, lower part). The prolonged drought event occurring before August 21/223 DOY contributed to spatial variability in RF rice as indicated by greater sill of 0.0142 compared with 0.0032 on August 08/220 DOY. ANOVA analysis indicated no difference at 0.05 significance level among PD three nutrient groups over the growing season (p = 0.67), whereas, mean sill value of 0.4492 on June 21/172 DOY was

improved by 93.32% for normal and high nutrient groups than 0.03 of low nutrient group, assembling comparisons in sill of $GPP_{day}$ and field maps shown in Fig. 6a. It implied that fertilizer addition could enhance spatial variability of $LUE_{cabs}$ especially early in growing seasons. Interestingly, at early growth stage especially on June 21/172 DOY and July 11/192, PD nutrient addition groups had averaged sill higher by approx. 85% as compared to RF rice. RF rice took over high values afterwards, meaning that spatial variability of $LUE_{cabs}$ in PD rice amplified by field nutrient application could be even greater

than RF rice, which totally contrasts with aforementioned $GPP_{day}$ spatial variability between PD and RF rice.

### 3.5 Spatial correlation for $GPP_{day}$, $LUE_{cabs}$, and LAI

$LUE_{cabs}$ was calculated by Eq. 8 consisting of $GPP_{day}$ and fAPR variables, meaning that spatial influences of $LUE_{cabs}$ may yield impacts on $GPP_{day}$. Sill values or $CV_{sill}$ for $GPP_{day}$ and $LUE_{cabs}$ were not significantly correlated to one another when all

data sets were grouped across PD nutrient groups and RF rice over growing seasons ($R^2 < 0.14$, $p > 0.01$). Instead, such significantly positive correlations were found for $sill_{GPPday}$-$sill_{LAI}$ in PD nutrient groups (Fig. 7d, $R^2 = 0.36$, $p = 0.012$) and in RF rice (Fig. 7d, $R^2 = 0.85$, $p = 0.015$), suggesting that the primary factor that mediates $GPP_{day}$ spatial variation in PD nutrient groups especially in RF rice was LAI development.

### 3.6 Imply ecological implications of canopy leaf physiology

Ecological implications of canopy leaf physiology i.e. $LUE_{cabs}$ in monitoring of spatial variation and strength of $GPP_{day}$ for the same plant function type (PD and RF rice) were analyzed using scenario analysis. It applied $LUE_{cabs}$ of PD rice on August 08/220 DOY in estimation of RF rice $GPP_{day}$ at the same date, yielding comparisons in field map of $GPP_{day}$ (Fig. 8a, b) and quantitative assessment (Fig. 8c). Field map of predicted $GPP_{day}$ using PD-$LUE_{cabs}$ indicated blue as prevailing color

as compared to prevailing red color in field map of initial estimation, meaning significant underestimations of $GPP_{day}$ especially at the sites where showed high LAI (Fig. 8c). It suggested that take delicate variations in canopy leaf physiology among the same plant function type across various habitat conditions into account seems to be vital.

### 4 Discussion

A series of successive effects regarding rice growth and environment from leaf to ecosystem perspectives has been made in our research group, aiming to unveil physiological mechanisms responsible for optimal carbon gain and water use at leaf level as well as their plastic acclimation to changing ambient environment (Xue et al., 2016b and c), disentangle roles of




canopy structure and function in determination of canopy carbon gain at individual organism subject to different field management methods and anthropogenic interventions (Lindner et al., 2016; Xue et al., 2016a), supplement understanding of climate change, phenology, and rice ecosystem photosynthetic productivity (Xue et al., 2017), and discuss ecological implications of life history of rice crop in controlling regional carbon fluxes at agriculture landscape (Lindner et al., 2015).

Great fluctuations of ecosystem photosynthetic productivity across different geographic sites existed. However, the fluctuation was not statistically correlated to nitrogen application rates which do significantly contribute to rice growth at individual level. It is thought to be due to various factors. At least, one of them could be ascribed to inter- and intra-field variations of ecosystem photosynthetic productivity, indicating that research specified into filed/microsite should be implemented to gain new insights into how water and nitrogen availability affect photosynthetic productivity at individual

and microsite scales.

**4.1 Feasible application of UAV system to capture spatiotemporal variations of GPP$_{day}$**

Applications of close-range remote sensing in studies of vegetation dynamics regarding plant growth and phenology have received increasingly concern partially due to pixel-to-pixel detection at small scale that eliminates the averaging involved in

larger pixels of satellite products. It compensates for regional observation of satellite remote sensing systems. UAV-based applications in agronomical studies has been tested, and evaluating spatial variability of soil nitrogen content in winter wheat field (Cao et al., 2012), detecting canopy nitrogen status in irrigated maize (Bausch and Khosla, 2010), and mapping cereal yield using field vegetation indices (VIs, Fisher et al., 2009; Swain et al., 2010; Tubaña et al., 2012; Zhang and Kovacs, 2012), rice growth and yield included (Ko et al., 2015). Recent attempts were made to apply narrow-band multispectral

imagery derived at plot level in monitoring of whole field carbon content of lucerne plants (Wehrhan et al., 2016). Furthermore, an applicable crop information delivery system tested in rice ecosystems by Ko et al. (2015) and Jeong et al. (2016), which takes several valuable VIs at high spatial resolution into account, well capture delicate changes in crop growth and yield among pixels. In this research, diagnostic information derived from images in high spatial resolution could be well linked to canopy biophysical traits in PD and RF rice, and draw seasonally zonal maps of GPP$_{day}$ and LUE$_{cabs}$ (Fig. 5 and 6),

and then assist in evaluation of spatial variation of those functional traits.

Practical application of the UAV technique in the field requires a number of procedural steps, including image pre-processing, image interpretation and data extraction. And integration of these data with agronomic data into expert systems still needs to be developed and improved before end products of remote sensing applications are taken into account by decision-making processes (Zhang and Kovacs, 2012). An empirical calibration method adopting spectral reflectance

from three types of PITs was applied to process radiometric correction, calibrate initially accessible UAV images on each measuring date. Although a close correspondence was commonly found between calibrated UAV reflectance and at-surface measurements at middle and late growing seasons, the empirical calibration tended to underestimate ground reflectance

especially in red reflectance at the early growth stage probably due to water scattering effects. UAV flight schedule always arranged at solar noon may not be the best option to obtain a close correspondence between camera reflectance and ground surface measurements at early growth stage. Another empirical regression linking calibrated UAV reflectance and plot measurements was applied instead of considering complex mechanisms of light scattering in area of physical category. The

methods used to recalibrate UAV images on June 21/172 DOY may yield biased estimation of field reflectance due to limited number of ground reflectance swatches that were deployed at limited space. Leaves of plants grown at fertilization addition conditions had enhanced nitrogen content at early growth stage, which directly contributes to greater $LUE_{cabs}$ (Sinclair and Horie, 1989; Xue et al., 2017). Whereas, $LUE_{cabs}$ at normal and high nutrient groups where plants accumulated more nitrogen in leaves on June 21 (Fig. 4a) calibrated on the basis of recalibrated UAV reflectance were averagely higher as

compared to low nutrient group (Table 2), which implies the pragmatic feasibility of adopting recalibration routine to acquire correct UAV products.

The data assimilation concept that integrates traditional physiology approaches at plot level and close-range remote sensing information requires reliable establishments regarding correlations between ground surface measurements of VIs and LAI, LAI and $LUE_{cint}$ and $GPP_{max}$. Reliable relationships between those biophysical traits were inferred across PD nutrient

groups and RF rice (Fig. 3). Nevertheless, there are limited data sets for LAI-$LUE_{cint}$ correlation in RF rice mainly due to labor deficits to intensively carry out measurements of diurnal courses of leaf and canopy gas exchange and measurements of other plant parameters in PD nutrient groups and RF rice. Supplementary data sets in terms of LAI-$LUE_{cint}$ correlation in RF rice as well as other main cops will be surely conducted when field conditions together with research fund are granted in near future.

### 4.2 Spatial variability of photosynthetic trait in RF field seems to be not always greater than PD field

There are continuously increasing water and food demands in rice as world population breaks through into a new record. Expand rice planting area over different geographic sites particularly in those regions lack of irrigation water resource and/or fundamental facility to flood fields and high possibility of occurrence of water scarcity in coming decades in flooded regions

have triggered increasing concerns associated with how water availability in RF field could influence spatiotemporal variations of ecosystem photosynthetic productivity as compared to PD field (Serraj et al., 2008). Spatial variations of $GPP_{day}$ and LAI in RF field were amplified compared to PD nutrient groups at corresponding growth stages (Table 3). However, spatial variation of $LUE_{cabs}$ at early growth stage (June 21/172 DOY and July 11/192 DOY) at PD fertilization groups was significantly greater than RF at the same time period, implying that spatial variability of photosynthetic trait in

RF field does not seem to be always higher than PD field depending on nutrient availability. We also found that nutrient addition at early growth stage could amplify spatial heterogeneity of $GPP_{day}$ and $LUE_{cabs}$ in PD field while, such nutritional effects dismissed at reproductive and ripening stages.



**4.3 Imply ecological implications of field niche in spatially hierarchical remote sensing network**

Better interpret ecosystem carbon dynamics in response to different field management methods and anthropogenic interventions via their influences on plant structure and physiology emphasizes the importance of *in situ* plot data. While plot

data provide our most detailed information on rice carbon and water gas exchange, applying this understanding to broader spatial and temporal domains requires scaling approaches. As aforementioned before, field niche which resides between *in situ* plot and regional dimension is supposed to be a key chain of spatially hierarchical remote sensing network (Masek et al., 2015; Pause et al., 2016). Applications of the data fusion at microsite/field scale that combine observations of *in situ* canopy structure and function with field crop information derived from the UAV system well capture critical growth information of

rice crop in space.

Spatial variations in $GPP_{day}$ over PD nutrient groups and RF rice tend to be primarily mediated by LAI. Canopy structure i.e. LAI is the main biotic factor in rice ecosystems that could yield large impacts in seasonal course of ecosystem photosynthetic productivity, which is in line with previous reports (Xue et al., 2017). Nevertheless, scenario analysis in Fig. 8 documented markedly underestimations of $GPP_{day}$ in RF rice at the beginning of ripening stage when apply $LUE_{cabs}$ of PD

rice in spatial monitoring of $GPP_{day}$ in RF field. Spatial fluctuations of daily GPP at ripening stage when canopy LAI maximizes could directly contribute to variations of overall growth season photosynthetic productivity in rice (Xue et al., 2017). Furthermore, enhanced $LUE_{cabs}$ in RF rice is suggested to be ascribed to improved nitrogen accumulation capacity after 180 DOY (Fig. 4), or due to phosphorus uptake efficiency (Kato et al., 2016) that was not quantified here. Changes in leaf nitrogen allocation within leaves that relate to photosynthetic activity of individual leaves may also have important

implications, i.e. improve plant biomass production (Karaba et al., 2007; Wang et al., 2014), visa verse, may not affect biomass (Tanaka et al., 2013; Dow and Bergmann, 2014), and must be investigated along with canopy structure. It includes important information that consider variations in canopy leaf physiology for the same plant function type across various habitat conditions essentially contributes to better monitoring of per-field photosynthetic productivity and biological interpretation of its spatial patterns using remote sensing technique.

**5 Conclusions**

As far as we know, this is the first work aiming to assess influences of nitrogen and water availability in spatial and temporal patterns of the rice ecosystem photosynthetic productivity at micro scale. Quantitatively abundant data at high quality derived from the close-range remote sensing system refract crop growth information linked to biotic and abiotic factors at

critical growth stages. Application of the data assimilation concept indicated that fertilizer addition in the PD rice field enhanced spatial variations of $GPP_{day}$ and LAI as well as $LUE_{cabs}$ at early growth stage. Change planting culture from flooded to rainfed conditions contributed to greater spatial heterogeneity of those traits. Nevertheless, nutritional effects in

the PD rice at early growth stage made PD field possess even greater spatial heterogeneity in $LUE_{cabs}$. Physiological basis related to $LUE_{cabs}$ in the RF rice highlighted that incorporate spatial variations of canopy leaf physiology for the same plant function type into field gas exchange modelling campaigns could substantially improve evaluation of ecosystem photosynthetic production at regional/continental scales.

**Appendices**

*Acknowledgement.* This study was supported by the Basic Science Research Program through the National Research Foundation of Korea (NRF), funded by the Ministry of Education, Science, and Technology (NRF-2013R1A2005788). We thank the agricultural logistics group of CNU for the field management. We do acknowledge the helps in the field by Steve Lindner, Bhone Nay-Htoon, Jinsil Choi, Seung Hyun Jo, Toncheng Fu, Fabian Fischer, Nikolas Lichtenwald and Yannic Ege. We gratefully acknowledge the technical assistance of Ms. Margarete Wartinger for all her support in the field and laboratory.

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



**Figure and Table**

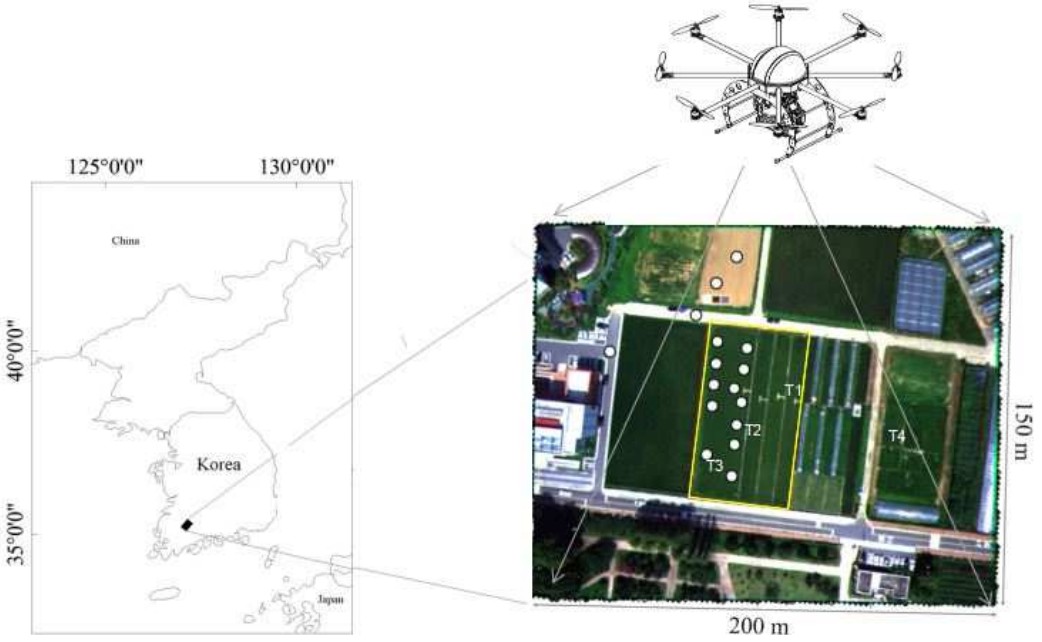

**Figure 1.** Illustration of study site where field data collection campaign that was carried out in 2013, Gwangju, S. Korea.

5      Yellow square and white circles represent sites of paddy fields and those marked for measurements of ground reflectance by

one handheld MSR to validate UAV imagery. T1: paddy rice under low nutrient condition (no supplementary nitrogen

applied); T2: PD rice under high nutrient condition (180 kg N ha$^{-1}$); T3: PD rice under normal nutrient condition (115 kg N

ha$^{-1}$), and T4: RF rice (115 kg N ha$^{-1}$). PD: paddy; RF: rainfed.





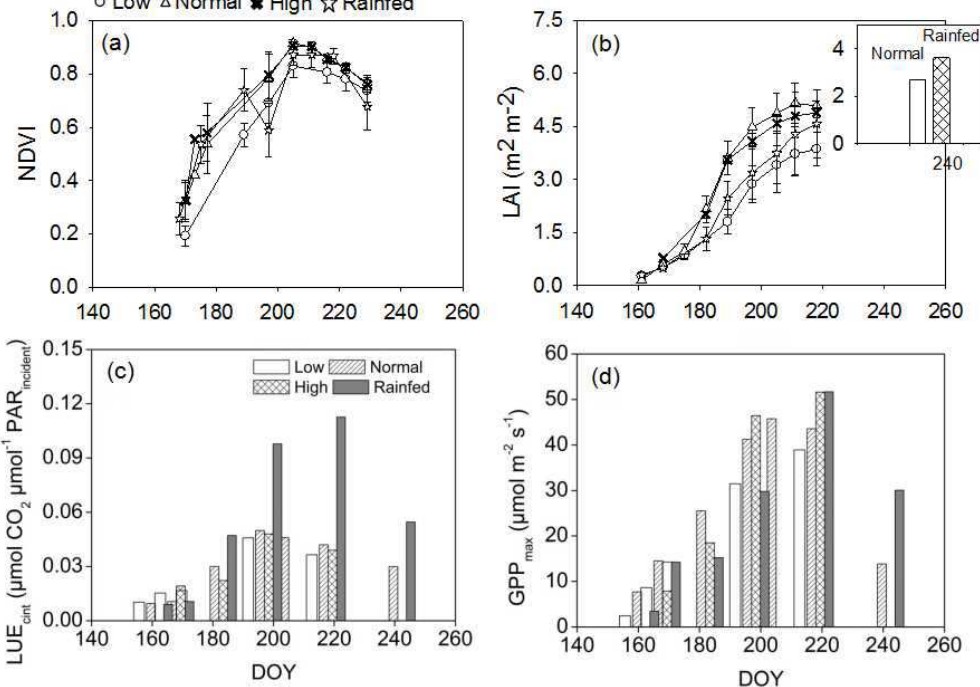

**Figure 2.** Seasonal courses of (a) normalized difference vegetation index (NDVI), (b) leaf area index (LAI), (c) canopy light use efficiency based on incident PAR ($LUE_{cint}$), and (d) maximum gross primary production ($GPP_{max}$) measured at plot level in PD low, normal and high nutrient groups, and in RF rice. Mean ± SD, n= 3 to 6. DOY: day of year. PD: paddy; RF: rainfed.





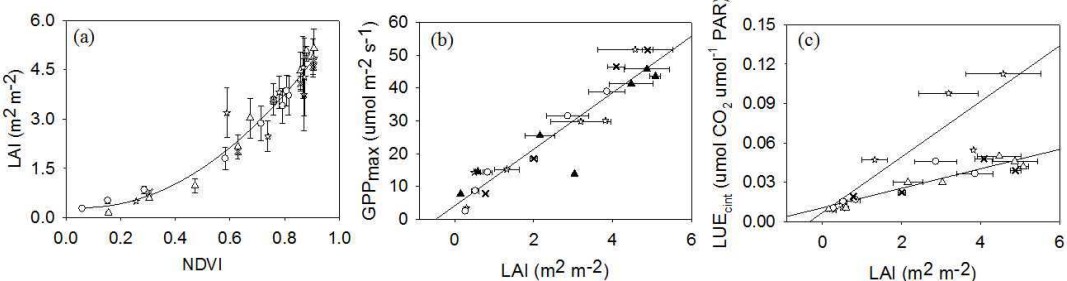

**Figure 3.** Correlations between (a) normalized difference vegetation index (NDVI) and leaf area index (LAI), (b) maximum

gross primary production (GPP$_{max}$) and LAI, and (c) canopy light use efficiency (LUE$_{can}$) and LAI across PD low, normal

and high nutrient groups, and in RF rice. Mean ± SD, n= 3 to 6. PD: paddy; RF: rainfed.



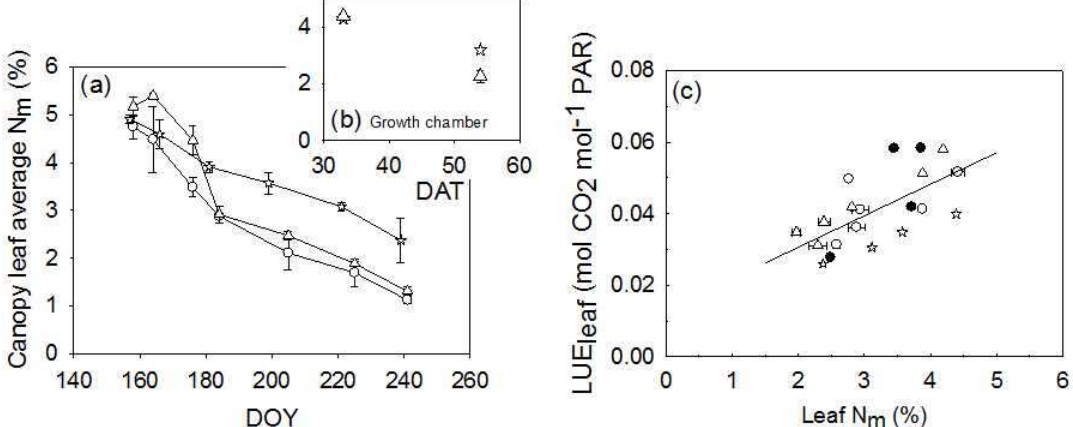

**Figure 4.** Seasonal development of leaf nitrogen content ($N_m$) in (a) PD low, normal and high nutrient groups, and in RF rice in the field, and (b) in PD and RF rice grown in controlled growth chamber. (c) Correlation between leaf light use efficiency ($LUE_{leaf}$) and $N_m$ crossing PD and RF rice. Mean ± SD, n= 3 to 6. DOY: day of year. DAT: day after transplanting. PD: paddy; RF: rainfed.





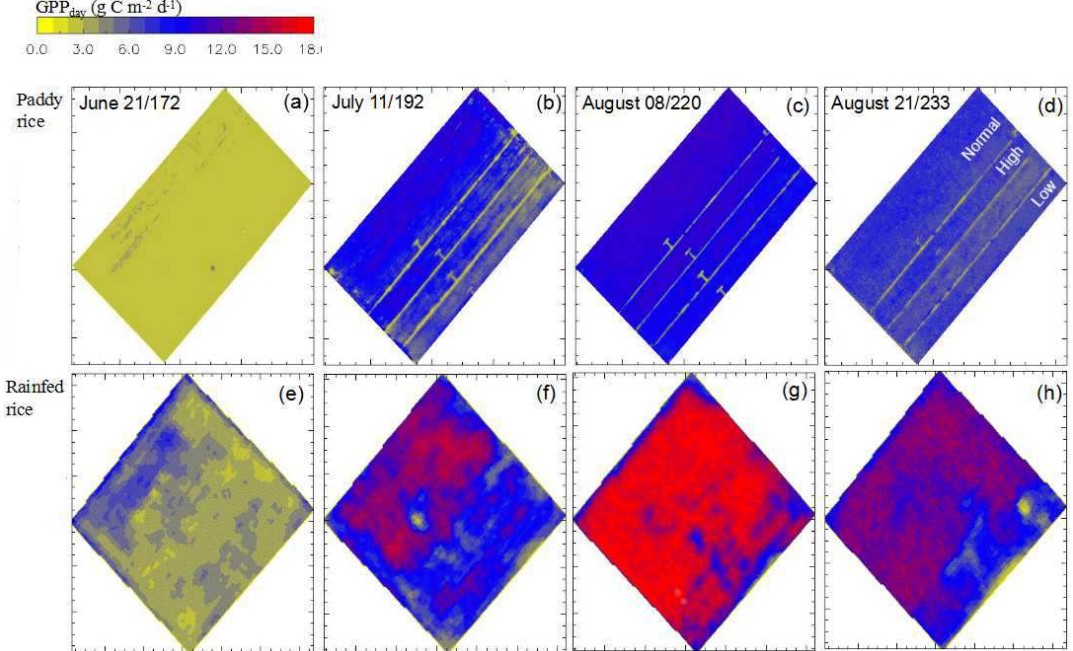

**Figure 5.** Filed mapping of ecosystem gross primary production (GPP) in PD rice and RF rice at principle growth stags: vegetative stage (June 21/172), middle reproductive stage (July 11/192), early ripening stage (August 08/220), and middle

5    ripening stage (August 21/233). Date ere expressed as MM DD/DOY. DOY: day of year; PD: paddy; RF: rainfed.

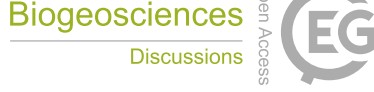

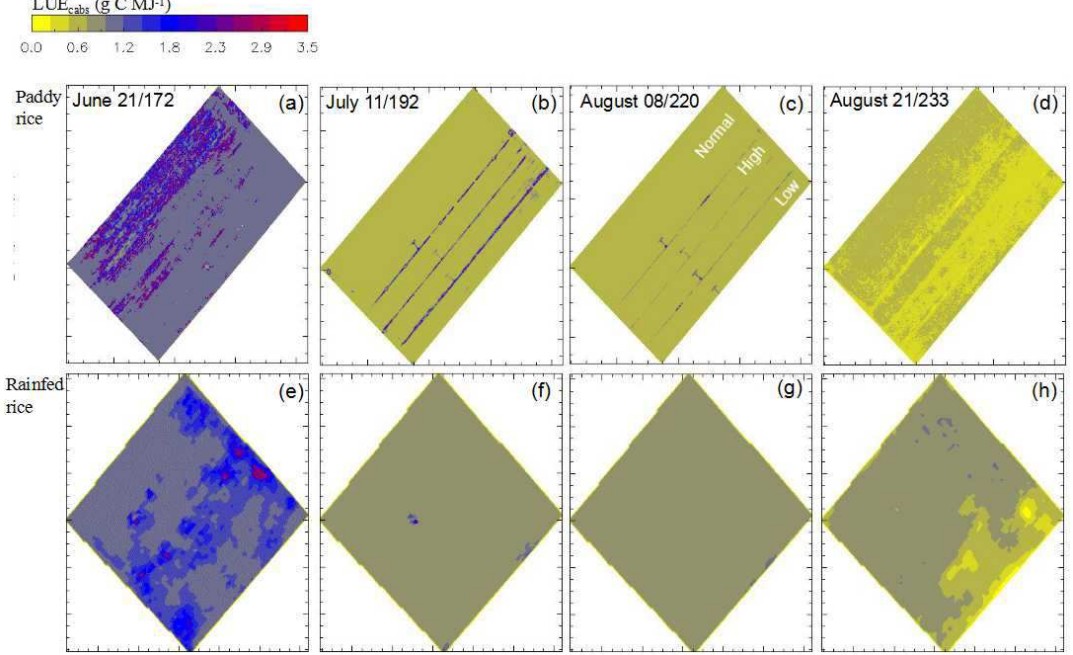

**Figure 6.** Filed mapping of canopy light use efficiency (LUE$_{cabs}$) in PD rice and RF rice at principle growth stags: vegetative

stage (June 21/172), middle reproductive stage (July 11/192), early ripening stage (August 08/220), and middle ripening

5   stage (August 21/233). Date ere expressed as MM DD/DOY. DOY: day of year; PD: paddy; RF: rainfed.





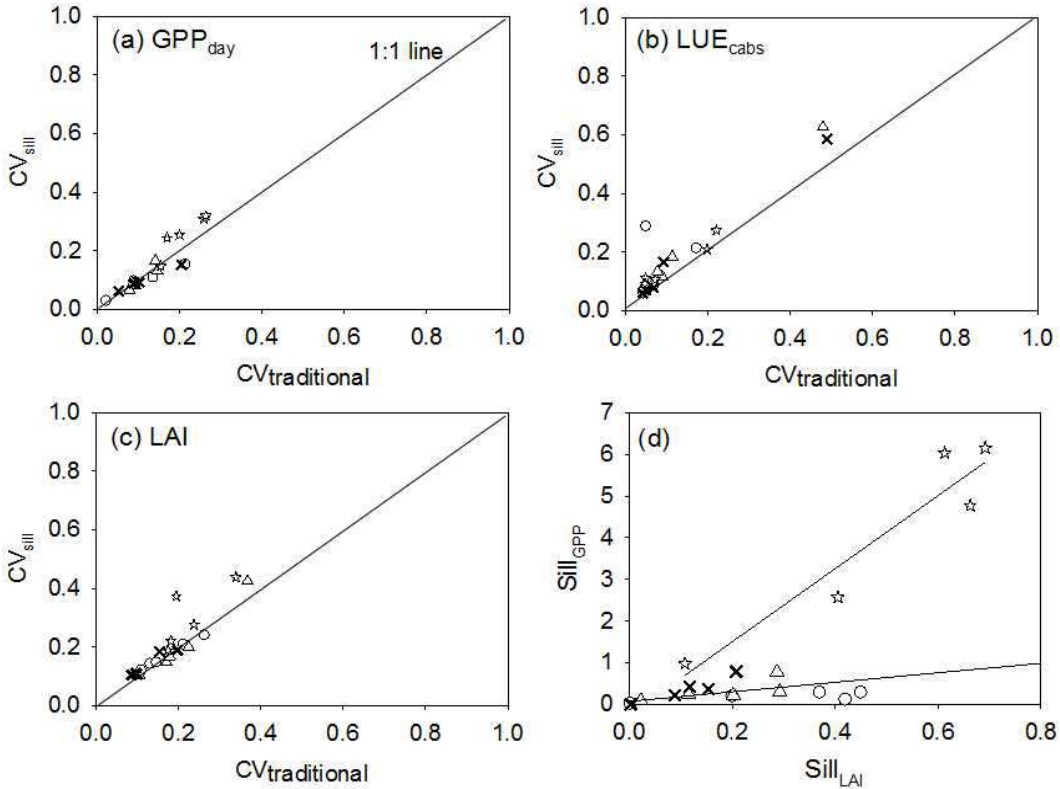

**Figure 7.** Coefficient of variation calculated by dividing the standard deviation by the mean ($CV_{traditional}$) versus coefficient of variation calculated using the semi-variogram sill ($CV_{sill}$) across PD nutrient groups and RF rice for variables (a) $GPP_{day}$, (b)

5    $LUE_{cabs}$, and (c) LAI. RF: rainfed; PD: paddy.



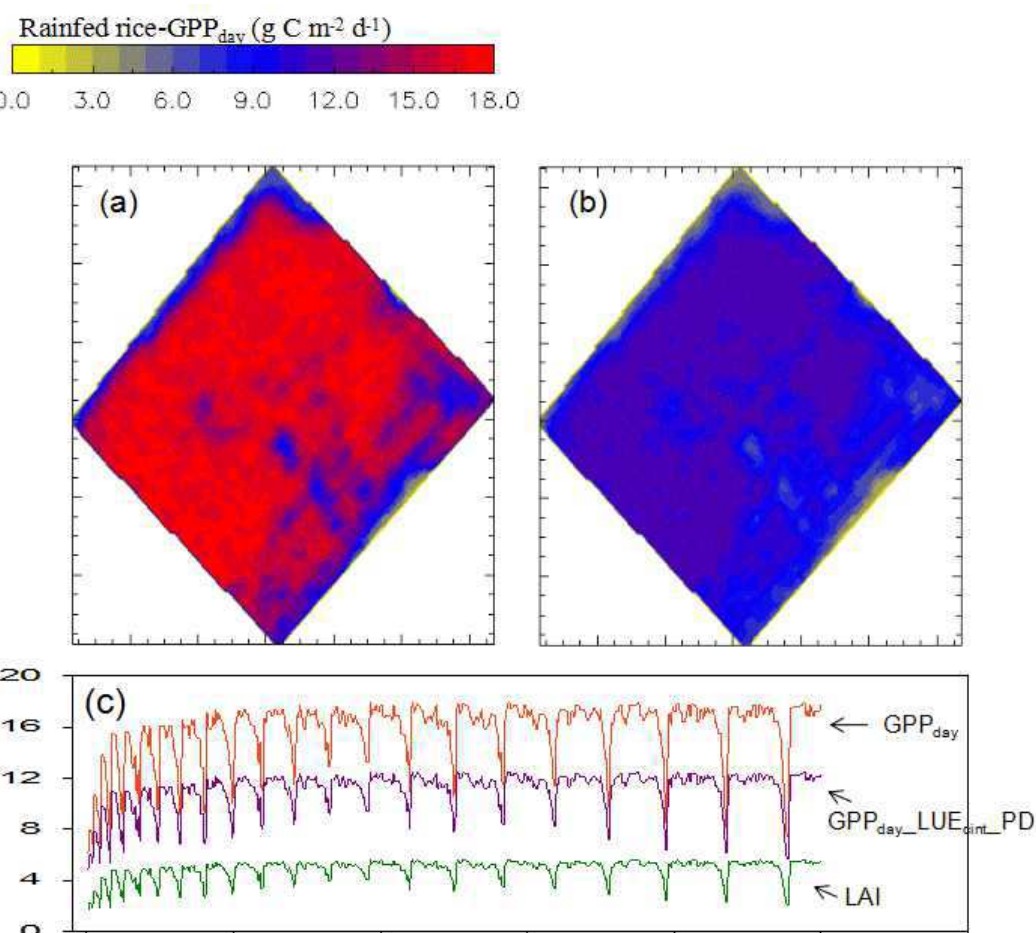

**Figure 8.** Evaluate potential effects of light use efficiency ($LUE_{cint}$) in ecosystem photosynthetic productivity ($GPP_{day}$) in field RF rice at ripening stage. $GPP_{day}$ estimation of RF rice was carried out by adopting $LUE_{cint}$ value of PD rice at ripening stage. $GPP_{day}$ estimation using (a) observed $LUE_{cint}$ in RF rice, (b) using $LUE_{cint}$ of PD rice ($GPP_{day\_}LUE_{cint\_}PD$), and (c) quantitative comparisons between $GPP_{day}$ and $GPP_{day\_}LUE_{cint\_}PD$ as referred to leaf area index (LAI). PD: paddy; RF: rainfed.





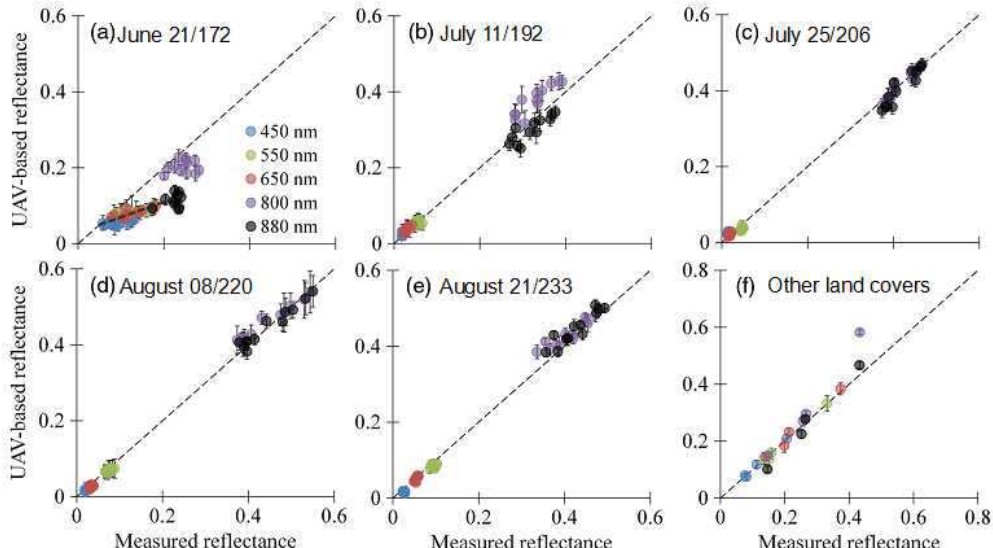

**Figure A1.** Validation of calibrated UAV-based reflectance by measurements of group point reflectance set up in paddy fields across the whole growing season (a-e) and in other land covers obtained on 172, 192, and 220 DOY including bright cement, dark asphalt, bare soil, and tilled soil (f). Dash line in each subplot shows 1:1 ratio. Recalibration for UAV-based reflectance in red waveband was conducted on June 21/172 DOY, shown in subplot a (coarse dash line). DOY: day of year.



**Table 1.** Values of coefficients for Eqs 3-7. PD: paddy rice; RF: rainfed rice.

| Eqs. | Coef. | Values | Coef. | Values | Coef. | Values | Coef. | Values |
|------|-------|--------|-------|--------|-------|--------|-------|--------|
| Eq. 2 | $a_2\_PD$ | 0.0074 | $b_2\_PD$ | 0.0107 | | | | |
| | $a_2\_RF$ | 0.0211 | $b_2\_RF$ | 0.0070 | | | | |
| Eq. 3 | $a_3$ | 8.571 | $b_3$ | 4.081 | | | | |
| Eq. 4 | $a_1$ | 7.398 | $b_1$ | -1.752 | $c_1$ | 0.452 | | |
| Eq. 6 | $fPAR_{max}$ | 0.95 | $NDVI_{max}$ | 0.94 | $NDVI_{min}$ | 0.11 | $\varepsilon$ | 0.6 |
| Eq. 7 | $a_4$ | 0.169 | $b_4$ | 0.765 | | | | |

* Values of coefficients for Eq. 7 were derived from reports by Inoue et al. (2008).





**Table 2.** Descriptive statistics of ecosystem photosynthetic productivity (GPP$_{day}$, g C m$^{-2}$ d$^{-1}$) and light use efficiency (LUE$_{cabs}$, g C MJ$^{-1}$) at each nutrient treatment in PD rice and at RF rice. Measuring date (MM DD/DOY). DOY: day of year. PD: paddy; RF: rainfed.

| GPP$_{day}$ | | | | LUE$_{cabs}$ | | | |
|---|---|---|---|---|---|---|---|
| Low | Normal | High | Rainfed | Low | Normal | High | Rainfed |
| June 21 /172 DOY | | | | | | | |
| Mean | 2.32 | 2.56 | 2.33 | 4.53 | 1.16 | 1.67 | 1.43 | 1.3 |
| Max. | 3.78 | 7.29 | 3.51 | 10.57 | ~3.50 | ~3.50 | ~3.50 | 3.18 |
| CV$_{traditional}$ | 2.16% | 14.06% | 5.15% | 25.81% | 17.24% | 47.90% | 48.95% | 22.00% |
| July 11 /192 DOY | | | | | | | |
| Mean | 6.16 | 9.57 | 8.35 | 10.99 | 0.68 | 0.62 | 0.73 | 0.86 |
| Max. | 11.21 | 12.73 | 11.97 | 16.93 | 1.72 | 2.75 | 2.86 | 2.35 |
| CV$_{traditional}$ | 21.36% | 14.52% | 20.37% | 26.32% | 4.92% | 11.29% | 9.20% | 7.09% |
| July 25 /206 DOY | | | | | | | |
| Mean | 7.93 | 9.74 | 9.45 | 14.28 | 0.7 | 0.68 | 0.68 | 1.08 |
| Max. | 10.97 | 11 | 11.04 | 17.15 | 0.87 | 1.32 | 1.06 | 1.79 |
| CV$_{traditional}$ | 13.55% | 9.22% | 10.12% | 16.89% | 4.38% | 8.82% | 4.94% | 4.81% |
| August 08 /220 DOY | | | | | | | |
| Mean | 9.56 | 10.85 | 10.57 | 15.41 | 0.66 | 0.62 | 0.63 | 0.87 |
| Max. | 12.28 | 12.49 | 12.41 | 18.11 | 1.58 | 1.42 | 1.57 | 0.95 |
| CV$_{traditional}$ | 8.89% | 7.77% | 8.77% | 15.36% | 4.54% | 4.19% | 4.12% | 4.65% |
| August 21 /233 DOY | | | | | | | |
| Mean | 7.13 | 7.69 | 7.45 | 12.14 | 0.49 | 0.52 | 0.52 | 0.81 |
| Max. | 9.94 | 10.73 | 10.22 | 15.91 | 0.66 | 0.71 | 0.68 | 1.05 |
| CV$_{traditional}$ | 9.23% | 8.49% | 8.88% | 19.91% | 6.93% | 7.69% | 6.73% | 19.75% |



**Table 3.** Sill values of semi-variograms and $CV_{sill}$ for $GPP_{day}$ (g C m$^{-2}$ d$^{-1}$, upper part), LAI (m$^2$ m$^{-2}$, middle part), and $LUE_{cabs}$ (g C MJ$^{-1}$, lower part) at PD rice subject to low, normal and high nutrient gradients and at RF rice over the growing seasons: vegetative stage (June 21), reproductive stage (July 11 and 25), ripening stage (August 08 and 21). DOY: day of year. PD: paddy; RF: rainfed.

| Growth stage | Date/DOY | Low | | Normal | | High | | Rainfed | |
|---|---|---|---|---|---|---|---|---|---|
| **$GPP_{day}$** | | Sill | $CV_{sill}$ | Sill | $CV_{sill}$ | Sill | $CV_{sill}$ | Sill | $CV_{sill}$ |
| Vegetative | June 21/172 | 0.01 | 2.86% | 0.09 | 16.57% | 0.01 | 6.10% | 0.98 | 30.91% |
| Reproductive | July 11/192 | 0.45 | 15.40% | 0.78 | 13.05% | 0.79 | 15.05% | 6.15 | 31.91% |
| | July 25/206 | 0.37 | 10.85% | 0.31 | 8.08% | 0.37 | 9.10% | 6.03 | 24.32% |
| Ripening | August 08/220 | 0.42 | 9.59% | 0.25 | 6.52% | 0.43 | 8.77% | 2.57 | 14.71% |
| | August 21/233 | 0.20 | 8.87% | 0.23 | 8.82% | 0.22 | 8.90% | 4.77 | 25.44% |
| **LAI** | | | | | | | | | |
| Vegetative | June 21/172 | 0.0015 | 14.19% | 0.0219 | 42.48% | 0.0026 | 18.40% | 0.1079 | 43.75% |
| Reproductive | July 11/192 | 0.1111 | 20.81% | 0.2869 | 19.96% | 0.2076 | 18.91% | 0.6915 | 37.36% |
| | July 25/206 | 0.1866 | 14.68% | 0.2924 | 14.76% | 0.1535 | 10.71% | 0.6127 | 22.07% |
| Ripening | August 08/220 | 0.4306 | 23.99% | 0.1148 | 10.10% | 0.1174 | 10.37% | 0.4050 | 18.83% |
| | August 21/233 | 0.0910 | 12.02% | 0.2015 | 16.72% | 0.0879 | 11.14% | 0.6622 | 27.59% |
| **$LUE_{cabs}$** | | | | | | | | | |
| Vegetative | June 21/172 | 0.0302 | 21.19% | 0.5478 | 62.68% | 0.3506 | 58.56% | 0.0633 | 27.37% |
| Reproductive | July 11/192 | 0.0190 | 28.67% | 0.0065 | 18.39% | 0.0073 | 16.55% | 0.0041 | 10.53% |
| | July 25/206 | 0.0008 | 5.71% | 0.0031 | 11.58% | 0.0011 | 6.90% | 0.0070 | 10.96% |
| Ripening | August 08/220 | 0.0011 | 7.11% | 0.0010 | 7.21% | 0.0007 | 5.94% | 0.0032 | 9.20% |
| | August 21/233 | 0.0009 | 8.66% | 0.0024 | 13.32% | 0.0008 | 7.69% | 0.0142 | 20.81% |