# Peer review of "Linking canopy reflectance to crop structure and photosynthesis to capture and interpret spatiotemporal dimensions of per-field photosynthetic productivity"

_Biogeosciences, 2016_

## Referee Comment (RC1) · Anonymous Referee #1 · 20 Dec 2016

General comments

I have reviewed the manuscript "Linking canopy reflectance to crop structure and photosynthesis to capture and interpret spatiotemporal dimensions of per-field photosynthetic productivity" (file: bg-2016-492-manuscript-version2.pdf) and find it suitable for publication in Biogeosciences. The presented research is interesting and relevant to the readership of Biogeosciences. The measurements and analyses appear to be sound. However, I would recommend thorough language editing as the manuscript contains grammatical errors and, occasionally, uncommon phrasings that make it a

bit difficult to understand the content at times. Apart from that, there are only some technical corrections to do, from my point of view.

Specific comments

P11, L9-10: "They imply that. . .", this sentence contains interpretations of the results, which I would rather avoid in the results chapter. The same applies to: P11, L18-19; P11, L 27-28; P13, L6; P13, L17-18.

Technical corrections

P9, L30: I would recommend to use the word "dramatically" here.

P11, L 2: "predicated" should read "predicted", I suppose.

Fig. 3: it would be helpful to add a legend of treatment symbols. In the y-axis labels, "umol" should read "$\mu$mol", I think. In the caption, it should read "gross primary productivity", in my understanding.

Fig. 4: add legend of treatment symbols.

Fig. 5: In the caption, it should read "gross primary productivity", "stages" (instead of "stags") and "are" (instead of "ere").

Fig. 6: "stages".

Fig. 7: item (d) is missing in the caption.

Fig. 8: x-axis label is missing.

---

## Referee Comment (RC2) · Anonymous Referee #2 · 27 Dec 2016

This paper, "Linking canopy reflectance to crop structure and photosynthesis to capture and interpret spatiotemporal dimensions of per-field photosynthetic productivity", is suitable for publication within Biogeosciences, however a number of questions and clarifications are necessary. The research is a unique effort to integrate multi-scale agricultural and ecophysiological measurements that readers would find interesting. Significant grammatical errors exist – if they were corrected then the paper would be much easier to digest and would more effectively convey its message.

Specific comments:

[Figure]

P2, L19: "... resolved using complex Bayesian melding". It was assumed that "melding" should be modeling; this reference doesn't provide evidence for why Bayesian hierarchical modeling would provide an adequate solution to solving the gaps in research.

Hypothesis 1: Reads as if was formulated after the research was completed, and should be composed in present tense. It would also be beneficial to explicitly separate the second part of this hypothesis into a new hypothesis (LAI and canopy leaf physiology as the primary(?) driver of spatial variation in GPP).

How many replications were used for each treatment? Were the treatments randomly assigned to plots? This is never addressed.

Top of page 5: It may be helpful to briefly mention why the measurement DOYs of the portable gas exchange and chlorophyll fluorescence systems did not match up and were not consistent across nutrient groups.

P4, L20-24. Need citations for these methods to estimate Reco and GPP. For equations 2 – 8, citations and/or greater justification for using the equations is warranted.

P7, L19, Eq 2: Need justification for LUE be a linear function of LAI (equation 2). The same holds for equation 3.

P7, L19, Eq 5: Based on this formulation, it appears as though LAI appears twice in the numerator – in the GPP and LUE terms from eq.2 and eq. 3. This should be justified. It would be helpful to have an explanation for why GPP is multiplied in the numerator and added in the denominator.

P8, lower half: Justify the use of a non-directional exponential semi-variogram. It would also be helpful to know the number of observations that are used to derive the semi-variogram (number of pixels?).

Was the semi-variogram applied to account for spatial autocorrelation of all the response variables or only some of them? This is detailed to some degree in the results

but should be explicitly stated here.

P9, L12. The difference between the PD high and normal groups may have been statistically significant, but was it practically significant (what was the magnitude?) Unclear whether difference between low group and mid/high PD groups is important.

Fig 2: Recommend breaking sub-figures a and b into two figures, one for PD, one for rainfed. It would also be helpful to have error bars on sub-figures c and d.

P10, L7. The distinction in LAI-LUE slopes should likely be tested with an interaction parameter and F-test rather than comparing R-squared and p-values.

P10, L18. Pink pixels in PD rice are not evident.

P12, L22. Was the paired t-test using observations across the range of DOY? This seems apparent in the subsequent text but should be stated up-front.

What was the background macro- and micro-nutrient concentrations in each treatment excluding N?

---

## Author Comment (AC1) · 12 Jan 2017

1. The presented research is interesting and relevant to the readership of Biogeosciences. The measurements and analyses appear to be sound. However, I would recommend thorough language editing as the manuscript contains grammatical errors and, occasionally, uncommon phrasings that make it a bit difficult to understand the content at times. Reply: A careful English edition on the precious version has been done. Please check the updated MS. 2. P11, L9-10: "They imply that: : :", this sentence contains interpretations of the results, which I would rather avoid in the

results chapter. The same applies to: P11, L18-19; P11, L 27-28; P13, L6; P13, L17-18. Reply: The word "imply" in results chapter was replaced by "suggest". 3. P9, L30: I would recommend to use the word "dramatically" here. Reply: Done 4. P11, L 2: "predicated" should read "predicted", I suppose. Reply: Done 5. Fig. 3: it would be helpful to add a legend of treatment symbols. In the y-axis labels, "umol" should read "$\mu$mol", I think. In the caption, it should read "gross primary productivity", in my understanding. Reply: Legend was added in Figs. 3, 4 and 7. Correct unit and terminology were added. 6. Fig. 4: add legend of treatment symbols. Reply: Legend was added in Figs. 3, 4 and 7. 7. Fig. 5: In the caption, it should read "gross primary productivity", "stages" (instead of "stags") and "are" (instead of "ere"). Reply: Correct terminology and word were added. 8. Fig. 6: "stages". Reply: Done 9. Fig. 7: item (d) is missing in the caption. Reply: Done 10. Fig. 8: x-axis label is missing. Reply: Done Thanks for your useful comments.

Please also note the supplement to this comment:
http://www.biogeosciences-discuss.net/bg-2016-492/bg-2016-492-AC1-supplement.pdf
* * *
[Figure]

**Supplement:**

[revised manuscript text omitted]
 regions that now feature flooding of crop fields is increasing the concerns of how water availability in RF fields could influence spatiotemporal variations of ecosystem photosynthetic productivity as compared to PD fields (Serraj et al., 2008). In the present study, spatial variations of $GPP_{day}$ and LAI in RF field were amplified compared to PD nutrient groups at corresponding growth stages (Table 3). However, spatial variation of $LUE_{cabs}$ at the early growth stage (June 21, 172 DOY and July 11, 192 DOY) in the PD fertilization groups was significantly greater than RF at the same times, suggesting that spatial variability of photosynthetic trait in RF field does not always exceed that of PD fields depending on nutrient availability. Furthermore, nutrient addition at the early growth stage could amplify spatial heterogeneity of $GPP_{day}$ and $LUE_{cabs}$ in PD field while, such nutritional effects dismissed at reproductive and ripening stages.

**4.3 Implied ecological implications of field niche in a spatially hierarchical remote sensing network**

*In situ* plot data is important for the more accurate interpretation of ecosystem C dynamics in response to different field

management methods and anthropogenic interventions that involve influences on plant structure and physiology. While plot data provides the most detailed information on rice C and water gas exchange, applying this understanding to broader spatial and temporal domains requires scaling approaches. As mentioned before, the field niche between *in situ* plot and regional dimension is supposed to be a key chain of a spatially hierarchical remote sensing network (Masek et al., 2015; Pause et al., 2016). Applications of the data fusion at the microsite/field scale that combine observations of *in situ* canopy structure and function with field crop information derived from the UAV system capture critical growth information of rice crop in space.

Spatial variations in $GPP_{day}$ over PD nutrient groups and RF rice tend to be primarily mediated by LAI. Canopy structure (i.e., LAI) is the main biotic factor in rice ecosystems that could have a great impact on the seasonal course of ecosystem photosynthetic productivity, consistent with previous reports (Xue et al., 2017). Nevertheless, the scenario analysis in Fig. 8 documented marked underestimations of $GPP_{day}$ in RF rice at the beginning of ripening stage when applying $LUE_{cabs}$ of PD rice in spatial monitoring of $GPP_{day}$ in the RF field. Spatial fluctuations of daily GPP at the ripening stage, when canopy LAI is maximized, could directly contribute to variations of overall growth season photosynthetic productivity in rice (Xue et al., 2017). Furthermore, enhanced $LUE_{cabs}$ in RF rice may reflect improved N accumulation capacity after 180 DOY (Fig. 4) or efficient P uptake (Kato et al., 2016) that we did not quantify. Changes in leaf N allocation within leaves that relate to photosynthetic activity of individual leaves may also have important implications like improved plant biomass production (Karaba et al., 2007; Wang et al., 2014) or may not affect biomass (Tanaka et al., 2013; Dow and Bergmann, 2014), and must be investigated along with canopy structure. Such investigations will need to consider variations in canopy leaf physiology for the same plant function type across various habitat conditions. The result will hopefully better monitoring of per-field photosynthetic productivity and biological interpretation of its spatial patterns using remote sensing technique.

**5 Conclusions**

As far as we know, this is the first work aiming to assess influences of N and water availability in spatial and temporal patterns of the rice ecosystem photosynthetic productivity at the micro scale. Abundant and high-quality data derived from the close-range remote sensing system refract crop growth information linked biotic and abiotic factors at critical growth stages. Application of the data assimilation concept indicated that fertilizer addition in the PD rice field enhanced spatial variations of $GPP_{day}$ and LAI as well as $LUE_{cabs}$ during early growth. Change planting culture from flooded to rainfed conditions contributed to greater spatial heterogeneity of those traits. Nevertheless, nutritional effects in the PD rice at the early growth stage produced greater spatial heterogeneity in $LUE_{cabs}$ in PD fields. Physiological basis related to $LUE_{cabs}$ in RF rice highlights 
[revised manuscript text omitted]

| Page 3: [1] Deleted | User | 02.01.2017 13:14:00 |
| --- | --- | --- |
| , to evaluate the following | | |

| Page 3: [1] Deleted | User | 02.01.2017 13:15:00 |
| --- | --- | --- |
| is:(1) T | | |

| Page 3: [1] Deleted | User | 02.01.2017 13:15:00 |
| --- | --- | --- |
| was | | |

| Page 3: [1] Deleted | User | 02.01.2017 13:15:00 |
| --- | --- | --- |
| driven | | |

| Page 3: [2] Deleted | User | 02.01.2017 13:16:00 |
| --- | --- | --- |
| Nevertheless, c | | |

| Page 3: [2] Deleted | User | 02.01.2017 13:16:00 |
| --- | --- | --- |
| one of | | |

| Page 3: [2] Deleted | User | 02.01.2017 13:16:00 |
| --- | --- | --- |
| s | | |

| Page 3: [2] Deleted | User | 02.01.2017 13:16:00 |
| --- | --- | --- |
| efficiency | | |

| Page 3: [2] Deleted | User | 02.01.2017 13:16:00 |
| --- | --- | --- |
| by | | |

| Page 3: [2] Deleted | User | 02.01.2017 13:18:00 |
| --- | --- | --- |
| GPP | | |

| Page 3: [3] Deleted | User | 02.01.2017 13:15:00 |
| --- | --- | --- |
| | | |

| Page 3: [3] Deleted | User | 02.01.2017 13:19:00 |
| --- | --- | --- |
| (2) S | | |

| Page 3: [3] Deleted | User | 02.01.2017 13:19:00 |
| --- | --- | --- |
| , and then g | | |

| Page 3: [3] Deleted | User | 02.01.2017 13:20:00 |
| --- | --- | --- |

(1) Temporal course of canopy carbon gain capacity was primarily driven by LAI development and solar radiation intensity at reproductive stage (Xue et al., 2016a; 2017). Nevertheless, canopy leaf physiology is one of primary factors that determine canopy light use efficiency and thereby carbon gain capacity (Sinclair and Horie, 1989). Hence, spatial variability of ecosystem GPP could be concurrently driven by canopy structure i.e. LAI and canopy leaf physiology i.e. $LUE_{cabs}$.

| Page 3: [4] Deleted | User | 02.01.2017 13:15:00 |
| --- | --- | --- |

(2) Shifts of planting culture from flooded to rainfed conditions mean that water availability tends to be a primary factor determining ecosystem photosynthetic productivity, and then growth of rainfed rice suffers from multiple uncertainties regarding timing/strength of precipitation and uptake of nutrient availability in soil (Kato et al., 2016). Significant changes in leaf and root anatomies, and canopy structure and function in rainfed field

could occur (Yoshida, 1981; Steudle, 2000). Greater variations in spatial aspects of ecosystem GPP, LAI and LUE$_{cabs}$ in rainfed lowland rice than flooded rice are therefore anticipated.

| Page 3: [5] Deleted | User | 02.01.2017 13:28:00 |
|---|---|---|
| F | | |

| Page 3: [5] Deleted | User | 02.01.2017 13:28:00 |
|---|---|---|
| . | | |

| Page 3: [5] Deleted | User | 02.01.2017 13:28:00 |
|---|---|---|
| ( | | |

| Page 3: [5] Deleted | User | 02.01.2017 13:28:00 |
|---|---|---|
| , | | |

| Page 3: [5] Deleted | User | 02.01.2017 13:28:00 |
|---|---|---|
| , | | |

| Page 3: [5] Deleted | User | 02.01.2017 13:28:00 |
|---|---|---|
| , | | |

| Page 3: [5] Deleted | User | 02.01.2017 13:29:00 |
|---|---|---|
| M | | |

| Page 3: [5] Deleted | User | 02.01.2017 13:29:00 |
|---|---|---|
| averaged | | |

| Page 3: [5] Deleted | User | 02.01.2017 13:29:00 |
|---|---|---|
| are approx. | | |

| Page 3: [5] Deleted | User | 02.01.2017 13:30:00 |
|---|---|---|
| is prevalent | | |

| Page 3: [5] Deleted | User | 02.01.2017 13:30:00 |
|---|---|---|
| fall | | |

| Page 3: [5] Deleted | User | 02.01.2017 13:31:00 |
|---|---|---|
| P | | |

| Page 3: [5] Deleted | User | 02.01.2017 13:31:00 |
|---|---|---|
| of | | |

| Page 3: [5] Deleted | User | 02.01.2017 13:31:00 |
|---|---|---|
| C | | |

| Page 3: [6] Deleted | Andreas | 06.01.2017 16:06:00 |
|---|---|---|
| m | | |

| Page 3: [6] Deleted | Andreas | 06.01.2017 16:05:00 |
|---|---|---|
| and P$_2$O$_5$ | | |

| Page 3: [7] Deleted | User | 02.01.2017 13:32:00 |
|---|---|---|
| ly | | |

| Page 3: [7] Deleted | User | 02.01.2017 13:32:00 |
|---|---|---|
| to | | |

| **Page 3: [7] Deleted** | **User** | **02.01.2017 13:32:00** |
| --- | --- | --- |

named paddy rice (PD)

| **Page 3: [7] Deleted** | **User** | **02.01.2017 13:32:00** |
| --- | --- | --- |

:

| **Page 3: [7] Deleted** | **User** | **02.01.2017 13:33:00** |
| --- | --- | --- |

:

| **Page 3: [7] Deleted** | **User** | **02.01.2017 13:33:00** |
| --- | --- | --- |

with

| **Page 3: [7] Deleted** | **User** | **02.01.2017 13:33:00** |
| --- | --- | --- |

was mixed

| **Page 3: [7] Deleted** | **User** | **02.01.2017 15:43:00** |
| --- | --- | --- |

three

| **Page 3: [7] Deleted** | **User** | **02.01.2017 15:43:00** |
| --- | --- | --- |

:

| **Page 3: [7] Deleted** | **User** | **02.01.2017 15:44:00** |
| --- | --- | --- |

,

| **Page 3: [7] Deleted** | **User** | **02.01.2017 15:44:00** |
| --- | --- | --- |

named

| **Page 3: [7] Deleted** | **User** | **02.01.2017 15:44:00** |
| --- | --- | --- |

,

| **Page 3: [7] Deleted** | **User** | **02.01.2017 15:44:00** |
| --- | --- | --- |

,

| **Page 3: [7] Deleted** | **User** | **02.01.2017 15:44:00** |
| --- | --- | --- |

N

| **Page 3: [7] Deleted** | **User** | **02.01.2017 15:44:00** |
| --- | --- | --- |

respectively

| **Page 3: [7] Deleted** | **User** | **02.01.2017 15:44:00** |
| --- | --- | --- |

width perimeter

| **Page 3: [7] Deleted** | **User** | **02.01.2017 15:45:00** |
| --- | --- | --- |

, inserted

| **Page 3: [7] Deleted** | **User** | **02.01.2017 15:45:00** |
| --- | --- | --- |

1 m depth

| **Page 3: [7] Deleted** | **User** | **02.01.2017 15:45:00** |
| --- | --- | --- |

nitrogen fertilizer was applied

| **Page 3: [7] Deleted** | **User** | **02.01.2017 15:46:00** |
| --- | --- | --- |

, and the rest used

| **Page 3: [7] Deleted** | **User** | **02.01.2017 15:47:00** |
| --- | --- | --- |

, and

| Page 3: [7] Deleted | User | 02.01.2017 15:47:00 |
|---|---|---|

and

| Page 3: [7] Deleted | User | 02.01.2017 15:55:00 |
|---|---|---|

,

| Page 3: [7] Deleted | User | 02.01.2017 15:55:00 |
|---|---|---|

with

| Page 3: [7] Deleted | User | 02.01.2017 15:56:00 |
|---|---|---|

as

| Page 3: [8] Deleted | User | 02.01.2017 15:56:00 |
|---|---|---|

conducted in

| Page 3: [8] Deleted | User | 02.01.2017 15:56:00 |
|---|---|---|

two times

| Page 3: [8] Deleted | User | 02.01.2017 15:57:00 |
|---|---|---|

No

| Page 3: [8] Deleted | User | 02.01.2017 15:57:00 |
|---|---|---|

ion was supplied at the RF field

| Page 11: [9] Deleted | User | 02.01.2017 17:27:00 |
|---|---|---|

much

| Page 11: [9] Deleted | User | 02.01.2017 17:27:00 |
|---|---|---|

by roughly 2 times

| Page 11: [9] Deleted | User | 02.01.2017 17:27:00 |
|---|---|---|

after

| Page 11: [9] Deleted | User | 02.01.2017 17:27:00 |
|---|---|---|

showed the

| Page 11: [9] Deleted | User | 02.01.2017 17:27:00 |
|---|---|---|

/

| Page 11: [9] Deleted | User | 02.01.2017 17:27:00 |
|---|---|---|

,

| Page 11: [9] Deleted | User | 02.01.2017 17:28:00 |
|---|---|---|

followed by

| Page 11: [9] Deleted | User | 02.01.2017 17:28:00 |
|---|---|---|

dismissed

| Page 11: [9] Deleted | User | 02.01.2017 17:28:00 |
|---|---|---|

which well aligns

| Page 11: [9] Deleted | User | 02.01.2017 17:28:00 |
|---|---|---|

u

| Page 11: [9] Deleted | User | 02.01.2017 17:28:00 |
|---|---|---|

y

| Page 11: [10] Deleted | User | 02.01.2017 17:29:00 |
|---|---|---|
| addition | | |

| Page 11: [10] Deleted | User | 02.01.2017 17:29:00 |
|---|---|---|
| at | | |

| Page 11: [10] Deleted | User | 02.01.2017 17:29:00 |
|---|---|---|
| could | | |

| Page 11: [10] Deleted | User | 02.01.2017 17:29:00 |
|---|---|---|
| s | | |

| Page 11: [10] Deleted | User | 02.01.2017 17:29:00 |
|---|---|---|
| in | | |

| Page 11: [11] Deleted | User | 02.01.2017 18:50:00 |
|---|---|---|
| correspondingly, | | |

| Page 11: [11] Deleted | User | 02.01.2017 18:50:00 |
|---|---|---|
| during which time | | |

| Page 11: [12] Deleted | User | 02.01.2017 18:51:00 |
|---|---|---|
| , | | |

| Page 11: [12] Deleted | User | 02.01.2017 18:51:00 |
|---|---|---|
| ed | | |

| Page 11: [12] Deleted | User | 02.01.2017 18:51:00 |
|---|---|---|
| / | | |

| Page 11: [12] Deleted | User | 02.01.2017 18:51:00 |
|---|---|---|
| / | | |

| Page 11: [12] Deleted | User | 02.01.2017 18:51:00 |
|---|---|---|
| . | | |

| Page 11: [12] Deleted | User | 02.01.2017 18:51:00 |
|---|---|---|
| , respectively | | |

| Page 11: [12] Deleted | User | 02.01.2017 18:51:00 |
|---|---|---|
| at | | |

| Page 11: [12] Deleted | User | 02.01.2017 18:52:00 |
|---|---|---|
| exerted markedly greater values by | | |

| Page 11: [12] Deleted | User | 02.01.2017 18:52:00 |
|---|---|---|
| . | | |

| Page 11: [12] Deleted | User | 02.01.2017 18:53:00 |
|---|---|---|
| / | | |

| Page 11: [12] Deleted | User | 02.01.2017 18:53:00 |
|---|---|---|
| / | | |

| Page 11: [12] Deleted | User | 02.01.2017 18:53:00 |
|---|---|---|
| / | | |

| Page 11: [12] Deleted | User | 02.01.2017 18:53:00 |
|---|---|---|
| ere | | |

| Page 11: [12] Deleted | User | 02.01.2017 18:53:00 |
|---|---|---|
| as well | | |

| Page 11: [12] Deleted | User | 02.01.2017 18:54:00 |
|---|---|---|
| / | | |

| Page 11: [12] Deleted | User | 02.01.2017 18:54:00 |
|---|---|---|
| / | | |

| Page 11: [12] Deleted | User | 02.01.2017 18:54:00 |
|---|---|---|
| well | | |

| Page 11: [13] Deleted | User | 02.01.2017 18:54:00 |
|---|---|---|
| | | |

| Page 11: [13] Deleted | User | 02.01.2017 18:55:00 |
|---|---|---|
| s | | |

| Page 11: [13] Deleted | User | 02.01.2017 18:55:00 |
|---|---|---|
| P | | |

| Page 11: [13] Deleted | User | 02.01.2017 18:55:00 |
|---|---|---|
| , | | |

| Page 13: [14] Deleted | User | 02.01.2017 19:02:00 |
|---|---|---|
| three | | |

| Page 13: [14] Deleted | User | 02.01.2017 19:02:00 |
|---|---|---|
| , | | |

| Page 13: [14] Deleted | User | 02.01.2017 19:05:00 |
|---|---|---|
| / | | |

| Page 13: [14] Deleted | User | 02.01.2017 19:06:00 |
|---|---|---|
| than | | |

| Page 13: [14] Deleted | User | 02.01.2017 19:06:00 |
|---|---|---|
| of | | |

| Page 13: [14] Deleted | User | 02.01.2017 19:06:00 |
|---|---|---|
| assembling | | |

| Page 13: [14] Deleted | User | 02.01.2017 19:06:00 |
|---|---|---|
| It | | |

| Page 13: [14] Deleted | User | 02.01.2017 19:06:00 |
|---|---|---|
| ould | | |

| Page 13: [15] Deleted | User | 02.01.2017 19:06:00 |
|---|---|---|
| s | | |

| Page 13: [15] Deleted | User | 02.01.2017 19:06:00 |
|---|---|---|
| / | | |

| Page 13: [15] Deleted | User | 02.01.2017 19:06:00 |
| / | | |

| Page 13: [15] Deleted | User | 02.01.2017 19:07:00 |
| d | | |

| Page 13: [15] Deleted | User | 02.01.2017 19:07:00 |
| higher by approx. | | |

| Page 13: [15] Deleted | User | 02.01.2017 19:07:00 |
| took over high values afterwards | | |

| Page 13: [15] Deleted | User | 02.01.2017 19:07:00 |
| which totally contrasts | | |

| Page 13: [16] Deleted | User | 02.01.2017 19:08:00 |
| yield impacts on | | |

| Page 13: [16] Deleted | User | 02.01.2017 19:26:00 |
| Instead, such s | | |

| Page 13: [16] Deleted | User | 02.01.2017 19:26:00 |
| , | | |

| Page 13: [16] Deleted | User | 02.01.2017 19:26:00 |
| , | | |

| Page 13: [17] Deleted | User | 02.01.2017 19:26:00 |
| It | | |

| Page 13: [17] Deleted | User | 02.01.2017 19:26:00 |
| / | | |

| Page 13: [17] Deleted | User | 02.01.2017 19:27:00 |
| at | | |

| Page 13: [17] Deleted | User | 02.01.2017 19:27:00 |
| F | | |

| Page 13: [17] Deleted | User | 02.01.2017 19:27:00 |
| meaning | | |

| Page 13: [17] Deleted | User | 02.01.2017 19:27:00 |
| s | | |

| Page 13: [17] Deleted | User | 02.01.2017 19:28:00 |
| where | | |

| Page 13: [17] Deleted | User | 02.01.2017 19:28:00 |
| ed | | |

| Page 13: [17] Deleted | User | 02.01.2017 19:28:00 |
| It | | |

| Page 13: [17] Deleted | User | 02.01.2017 19:28:00 |
| ed | | |

| Page 13: [17] Deleted | User | 02.01.2017 19:28:00 |
|---|---|---|

take

| Page 13: [18] Deleted | User | 02.01.2017 19:33:00 |
|---|---|---|

perspectives has been made in

| Page 13: [18] Deleted | User | 02.01.2017 19:34:00 |
|---|---|---|

ing to unveil

| Page 13: [18] Deleted | User | 02.01.2017 19:34:00 |
|---|---|---|

disentangle

| Page 13: [18] Deleted | User | 02.01.2017 19:35:00 |
|---|---|---|

ation of

| Page 13: [18] Deleted | User | 02.01.2017 19:35:00 |
|---|---|---|

at

| Page 13: [18] Deleted | User | 02.01.2017 19:35:00 |
|---|---|---|

subject to

| Page 13: [18] Deleted | User | 02.01.2017 19:35:00 |
|---|---|---|

methods

| Page 14: [19] Deleted | User | 02.01.2017 19:35:00 |
|---|---|---|

supplement

| Page 14: [19] Deleted | User | 02.01.2017 19:36:00 |
|---|---|---|

at

| Page 14: [19] Deleted | User | 02.01.2017 19:36:00 |
|---|---|---|

Great

| Page 14: [19] Deleted | User | 02.01.2017 19:36:00 |
|---|---|---|

across

| Page 14: [19] Deleted | User | 02.01.2017 19:36:00 |
|---|---|---|

existed

| Page 14: [19] Deleted | User | 02.01.2017 19:36:00 |
|---|---|---|

was not

| Page 14: [19] Deleted | User | 02.01.2017 19:37:00 |
|---|---|---|

nitrogen

| Page 14: [19] Deleted | User | 02.01.2017 19:37:00 |
|---|---|---|

rates

| Page 14: [19] Deleted | User | 02.01.2017 19:40:00 |
|---|---|---|

It

| Page 14: [19] Deleted | User | 02.01.2017 19:40:00 |
|---|---|---|

. At least, one of them could be ascribed to

| Page 14: [19] Deleted | User | 02.01.2017 19:41:00 |
|---|---|---|

, indicating that research specified into

| Page 14: [19] Deleted | User | 02.01.2017 19:41:00 |

l

| Page 14: [19] Deleted | User | 02.01.2017 19:41:00 |

should be implemented

| Page 14: [19] Deleted | User | 02.01.2017 19:41:00 |

nitrogen

| Page 14: [20] Deleted | User | 02.01.2017 19:42:00 |

received increasingly concern

| Page 14: [20] Deleted | User | 02.01.2017 19:44:00 |

at small scale

| Page 14: [20] Deleted | User | 02.01.2017 19:44:00 |

evaluating

| Page 14: [20] Deleted | User | 02.01.2017 19:44:00 |

nitrogen

| Page 14: [20] Deleted | User | 02.01.2017 19:45:00 |

ng

| Page 14: [20] Deleted | User | 02.01.2017 19:45:00 |

nitrogen

| Page 14: [20] Deleted | User | 02.01.2017 19:46:00 |

 (Vis,

| Page 14: [20] Deleted | User | 02.01.2017 19:46:00 |

,

| Page 14: [20] Deleted | User | 02.01.2017 19:46:00 |

carbon

| Page 14: [20] Deleted | User | 02.01.2017 19:47:00 |

takes

| Page 14: [20] Deleted | User | 02.01.2017 19:47:00 |

VIs

| Page 14: [20] Deleted | User | 02.01.2017 19:47:00 |

at high spatial resolution

| Page 14: [20] Deleted | User | 02.01.2017 19:47:00 |

well

| Page 14: [20] Deleted | User | 02.01.2017 19:47:00 |

 images in

| Page 14: [20] Deleted | User | 02.01.2017 19:47:00 |

well

| Page 14: [20] Deleted | User | 02.01.2017 19:48:00 |

and draw

| Page 14: [20] Deleted | User | 02.01.2017 19:48:00 |
|---|---|---|

then

| Page 14: [21] Deleted | User | 03.01.2017 09:32:00 |
|---|---|---|

, including

| Page 14: [21] Deleted | User | 03.01.2017 09:32:00 |
|---|---|---|

And i

| Page 14: [21] Deleted | User | 03.01.2017 09:32:00 |
|---|---|---|

se

| Page 14: [21] Deleted | User | 03.01.2017 09:32:00 |
|---|---|---|

to

| Page 14: [21] Deleted | User | 03.01.2017 09:33:00 |
|---|---|---|

are taken into account by

| Page 14: [21] Deleted | User | 03.01.2017 09:33:00 |
|---|---|---|

processes

| Page 14: [21] Deleted | User | 03.01.2017 09:35:00 |
|---|---|---|

,

| Page 14: [21] Deleted | User | 03.01.2017 09:35:00 |
|---|---|---|

initially

| Page 14: [21] Deleted | User | 03.01.2017 09:35:00 |
|---|---|---|

a close correspondence was commonly found between

| Page 14: [21] Deleted | User | 03.01.2017 09:36:00 |
|---|---|---|

at

| Page 14: [21] Deleted | User | 03.01.2017 09:36:00 |
|---|---|---|

arranged

| Page 15: [22] Deleted | User | 03.01.2017 09:49:00 |
|---|---|---|

there are limited data sets for

| Page 15: [23] Deleted | User | 03.01.2017 09:49:00 |
|---|---|---|

labor deficits to intensively carry out

| Page 15: [24] Deleted | User | 03.01.2017 09:50:00 |
|---|---|---|

when field conditions together with research fund are granted in near future

| Page 15: [25] Deleted | User | 03.01.2017 09:51:00 |
|---|---|---|

re are continuously increasing

| Page 15: [26] Deleted | User | 03.01.2017 09:51:00 |
|---|---|---|

demands in rice as world population breaks through into a new record

| Page 15: [27] Deleted | User | 03.01.2017 09:53:00 |
|---|---|---|

fundamental facility to flood fields

| Page 15: [28] Deleted | User | 03.01.2017 09:55:00 |
|---|---|---|

does not seem to be always higher than

made PD field possess even

---

## Author Comment (AC2) · 12 Jan 2017

Reply to Referee#2

1. The research is a unique effort to integrate multi-scale agricultural and ecophysiological measurements that readers would find interesting. Significant grammatical errors exist – if they were corrected then the paper would be much easier to digest and would more effectively convey its message.

**Reply:** Grammatical errors in text body were corrected. Please kindly check the updated version.

2. P2, L19: ": : : resolved using complex Bayesian melding". It was assumed that "melding" should be modeling; this reference doesn't provide evidence for why Bayesian hierarchical modeling would provide an adequate solution to solving the gaps in research.

**Reply:** That reference provides a possible technical routine to cover remote sensing hierarchical network. The previous sentence that discusses Bayesian melding was removed.

3. Hypothesis 1: Reads as if was formulated after the research was completed, and should be composed in present tense. It would also be beneficial to explicitly separate the second part of this hypothesis into a new hypothesis (LAI and canopy leaf physiology as the primary(?) driver of spatial variation in GPP).

**Reply:** Yes, it is well formulated after the field research completion in 2013 but a general framework associated with up-scaling estimation of crop photosynthetic productivity was planned in 2012 before commenced field experiments. That is why measurements at multi-spatial and temporal scales were deployed. Darwin never knew what is species change and evolution before doing lots of species investigation. We think it's not the matter when the hypothesis is proposed before or after research completion, the valuable properties of them rely on which kind of scientific question could be potentially related and resolved.
The two hypotheses were re-organized in the improved MS.

4. How many replications were used for each treatment? Were the treatments randomly assigned to plots? This is never addressed.

**Reply:** At least three replications were collected for each data sets used for statistic analysis. Please kindly check the description in Materials and Methods chapter. Healthy plants were randomly selected for gas exchange measurements. Please refer to pervious publications that described the same field experiments as in this MS (Lindner et al., 2016; Xue et al., 2017).

5. Top of page 5: It may be helpful to briefly mention why the measurement DOYs of the portable gas exchange and chlorophyll fluorescence systems did not match up and were not consistent across nutrient groups.

**Reply:** Measurements of diurnal courses of leaf gas exchange across PD nutrient treatments and RF rice over the growing seasons were intensively conducted. It is impossible to arrange such measurements for three nutrient groups on the same day because we only have one GFS photosynthesis system. Please refer to pervious publications that described the same field experiments as in this MS (Xue et al., 2016a, b, c).

6. P4, L20-24. Need citations for these methods to estimate Reco and GPP. For equations 2 – 8, citations and/or greater justification for using the equations is warranted.

**Reply:** Citations for these methods to estimate Reco and GPP were supplemented in P4 L20-24 (Xue et al., 2016a; Lindner et al., 2016). For equations 2 – 8, citations that show key parameters were already cited in P7 L25. Please kindly check them.

7. P7, L19, Eq 2: Need justification for LUE be a linear function of LAI (equation 2). The same holds for equation 3.

**Reply:** The findings here in terms of $LUE_{cint}$-LAI and GPPmax-LAI were consistent with previous reports (Lindner et al., 2015; 2016). The relevant citations were added in P7 L19.

8. P7, L19, Eq 5: Based on this formulation, it appears as though LAI appears twice in the numerator – in the GPP and LUE terms from eq.2 and eq. 3. This should be justified.

It would be helpful to have an explanation for why GPP is multiplied in the numerator and added in the denominator.

**Reply:** GPPmax and LUEcint are maximum gross primary productivity at relatively infinite high PAR and light use efficiency based on incident PAR, respectively, two key parameters comprising a classic photosynthesis light response model (Eq.1 in Owen et al., 2007, and Eq.2 in Lindner et al., 2015), see below,

$GPP = \dfrac{\alpha \times \beta \times PAR}{\alpha \times PAR + \beta}$ , where α here is LUEcint, and β is GPPmax.

Use LAI may bring some biased estimations of α and β. Those two parameters tend to parallel change over growing seasons, which correlate to LAI, consistent with our previous reports.

9. P8, lower half: Justify the use of a non-directional exponential semi-variogram. It would also be helpful to know the number of observations that are used to derive the semi-variogram (number of pixels?). Was the semi-variogram applied to account for spatial autocorrelation of all the response variables or only some of them? This is detailed to some degree in the results but should be explicitly stated here.

**Reply:** All data sets/pixels (around 41536 pixels for high group, 124236 pixels for normal group, 39928 pixels for low group, and 6860 pixels for rainfed rice) in each nutrient and water treatment were processed in the calculation of semi-variogram, which run in an IDL/ENVI

integrated environment and looked for an overall pattern between proximity and the similarity of pixel values (P8 L19-20).

10. P9, L12. The difference between the PD high and normal groups may have been statistically significant, but was it practically significant (what was the magnitude?) Unclear whether difference between low group and mid/high PD groups is important.

**Reply:** Statistic analysis in leaf and canopy photosynthetic traits as well as biomass production between PD normal and high groups showed that there was no statistically significant difference. But it may not persist when take variations within fields into account. Variations in photosynthetic productivity in PD high nutrient field may make eigenvalue distinctly greater as compared to PD normal field at specific growth stage. This may also occur when compare within-field functional traits between PD low and normal, RF rice. We highlighted those in Results part.

Those photosynthetic traits on which biomass production tightly depends could be quantified by at-surface traditional physiological measurements, and are directly relevant for hierarchical remote sensing network.

11. Fig 2: Recommend breaking sub-figures a and b into two figures, one for PD, one for rainfed. It would also be helpful to have error bars on sub-figures c and d.

**Reply:** Error bars on sub-figures c and d has been provided. We suggest to group those four subplots in one figure as we have now.

12. P10, L7. The distinction in LAI-LUE slopes should likely be tested with an interaction parameter and F-test rather than comparing R-squared and p-values.

**Reply:** Comparing two regression line was done in R, showing significant difference at 0.01 level in slope of regression line (P10, L4).

13. P10, L18. Pink pixels in PD rice are not evident.

**Reply:** There is indeed pink colors, a product of blue and red, shown in Fig. 5c. Please zoom in Fig. 5c in a relative advanced computer system 64-bit with high screen resolution (for example 1920 * 1080). An aesthetical Fig. 5c was reproduced, see below,

[Figure]

Fig.5c Field map of GPP$_{day}$ in paddy rice.

14. P12, L22. Was the paired t-test using observations across the range of DOY? This seems apparent in the subsequent text but should be stated up-front.

**Reply:** This is stated up-front (P12, L20).

15. What was the background macro- and micro-nutrient concentrations in each treatment excluding N?

**Reply:** Description of study site including soil information was stated in Materials and Methods (P3, L20-21). Other nutrient elements that may bring large effects on plant growth and development were discussed in P16, L14.

Reference:

Owen, K.E., Tenhunen, J., et al. Linking flux network measurements to continental scale simulations: Ecosystem carbon dioxide exchange capacity under non-water-stressed conditions. Global Change Biology, 13, 734-760.

Lindner, S., Otieno, D., Lee, B., Xue, W., Arnhold, S., Kwon, H., Huwe, B. and Tenhunen, J.: Carbon dioxide exchange and its regulation in the main agro-ecosystems of Haean catchment in South Korea, Agr. Ecosyst. Environ., 199, 132-145, 2015.

Lindner, S., Xue, W., Nay-Htoon, B., Choi, J., Ege, Y., Lichtenwald, N., Fischer, F., Ko, J., Tenhunen, J. and Otieno, D.: Canopy scale $CO_2$ exchange and productivity of

transplanted paddy and direct seeded rainfed rice production systems in S. Korea, Agr. Forest Meteorol., 228, 229-238, 2016.

Xue, W., Lindner, S., Dubbert, M., Otieno, D., Ko, J., Muraoka, H., Werner, C. and Tenhunen, J.: Supplement understanding of the relative importance of biophysical factors in determination of photosynthetic capacity and photosynthetic productivity in rice ecosystems, Agr. Forest Meteorol., 232, 550-565, 2017.

Xue, W., Lindner, S., Nay-Htoon, B., Dubbert, M., Otieno, D., Ko, J., Muraoka, H., Werner, C., Tenhunen, J. and Harley, P.: Nutritional and developmental influences on components of rice crop light use efficiency, Agr. Forest Meteorol., 223, 1-16, 2016a.

Xue, W., Nay-Htoon, B., Lindner, S., Dubbert, M., Otieno, D., Ko, J., Werner, C. and Tenhunen, J.: Soil water availability and capacity of nitrogen accumulation influence variations of intrinsic water use efficiency in rice, J. Plant Physiol., 193, 26-36, 2016b.

Xue, W., Otieno, D., Ko, J., Werner, C. and Tenhunen, J.: Conditional variations in temperature response of photosynthesis, mesophyll and stomatal control of water use in rice and winter wheat, Field Crop. Res., 199, 77-88, 2016c.

---

## Author Response (AR1)

Reply to ACD,

1. I agree with the referees that the manuscript is interesting and timely, and also that English usage challenges mask its full potential. The revised manuscript is an improvement, but the manuscript would deliver its message more effectively if the wording was further clarified and simplified. The first sentence is an example. It states, "Nitrogen and water availability are two staple environmental elements in agroecosystems that can substantially alter canopy structure and physiology then crop growth, yielding large impacts on ecosystem regulating/production provisions." How about instead, "Nitrogen and water availability alter canopy structure and physiology - and thereby crop growth and yield - in agroecosystems." This could be revised further, but says the same thing in 10 fewer words. Note that this is one of many examples throughout the manuscript where text could be made more efficient; most sentences should be revised. I personally found the Discussion to be interesting and insightful, noting the critical role of further text edits. Please continue revising the manuscript for clarity, after which I will not hesitate to recommend the manuscript be published in Biogeosciences.

**Reply:** A careful edition throughout text body has been done. Please check the revisions that are highlighted in red color.

2. On line 8, I think of a data assimilation (framework) as one in which a model and measurement are formally fused; the model isn't apparent in the subsequent statement or manuscript. Because I don't see any application of a formal data assimilation procedure (e.g. via the Kalman Filter or other Bayesian techniques), I recommend choosing another term. Simply linking remote sensing with leaf and plant-scale models isn't formal data assimilation, it's parameter fitting, which of course is fine.

**Reply:** We respect your suggestions, and already replaced data assimilation by data filtering.

3. On line 12, 'available nutrient availability'? A substantial amount of editing remains to be completed.

**Reply:** three nitrogen application rates were used to replace the previous expression.

4. 'amplified' on line 14 is qualitative. By how much? Quantitative abstracts are more effective.

**Reply:** The word 'enlarged' was used there

5. Please use the multiplication sign instead of the star on page 6 and elsewhere, noting page 7 as an excellent example of proper usage.

**Reply:** That was modified in line 30 on page 6.

6. The use of 'low', 'normal', and 'high' in the results could be made more clear to the reader, perhaps by using a table.

**Reply:** For sake of brevity, the nomination of 'low', 'normal', and 'high' what are firstly denoted in Study site is proper in Results part.

7. 51.60 micromoles of $CO_2$ per m2 per second is perhaps too many significant digits. (See also '35.63% enhanced field mean of GPPday' below, and elsewhere.)

**Reply:** Two-digit format in percentage expression was adopted. But four digits in numerical value such as 51.60 micromoles of $CO_2$ per m2 per second were retained in text body. These could also be changed if you stick to your opnions.

8. On page 9, which 'classical light response model'? In my opinion there are many.

**Reply:** A proper citation Eq. 5 was inserted there.

9. Page 11 line 5 use of 'then'.

**Reply:** Done

Thanks a lot for your kind care. Please let us know without ant hesitation if there are any problems.

[revised manuscript text omitted]

---

## Author Response (AR2)

Reply to Editor,

The authors curiously ignored my suggestion regarding significant digits throughout much of the text and tables, which does not arise from opinion, rather years of experience in measuring these and similar variables. Not only is it more accurate to list a representative amount of significant digits given measurement uncertainty, it is also easier for the reader to comprehend. The digital quality of some figures (e.g. Figure 2) is also lacking, at least in the built pdf, and Figure 4b could be improved for clarity. Please make these technical corrections and any additional steps that will enhance the readability of the manuscript, after which I will recommend it be accepted in Biogeosciences.

**Reply:** Statistic analysis was added in text and Figs. 2 and 4b.  Fig. 4b was clarified. Please kindly check the uploaded MS.

Please let us know if there are any problems without any hesitations.